# Highly Valuable Polyunsaturated Fatty Acids from Microalgae: Strategies to Improve Their Yields and Their Potential Exploitation in Aquaculture

**DOI:** 10.3390/molecules26247697

**Published:** 2021-12-20

**Authors:** Anna Santin, Monia Teresa Russo, Maria Immacolata Ferrante, Sergio Balzano, Ida Orefice, Angela Sardo

**Affiliations:** 1Stazione Zoologica Anton Dohrn, Villa Comunale, 80121 Naples, Italy; anna.santin@szn.it (A.S.); monia.russo@szn.it (M.T.R.); sergio.balzano@szn.it (S.B.); ida.orefice@szn.it (I.O.); 2Department of Marine Microbiology and Biogeochemistry, Netherland Institute for Sea Research, Landsdiep 4, 1793 AB Texel, The Netherlands; 3Istituto di Scienze Applicate e Sistemi Intelligenti “Eduardo Caianiello”, Via Campi Flegrei 34, 80078 Pozzuoli, Italy

**Keywords:** microalgae, polyunsaturated fatty acids (PUFAs), genetic engineering, growth conditions, marketable PUFAs, sustainable processes

## Abstract

Microalgae have a great potential for the production of healthy food and feed supplements. Their ability to convert carbon into high-value compounds and to be cultured in large scale without interfering with crop cultivation makes these photosynthetic microorganisms promising for the sustainable production of lipids. In particular, microalgae represent an alternative source of polyunsaturated fatty acids (PUFAs), whose consumption is related to various health benefits for humans and animals. In recent years, several strategies to improve PUFAs’ production in microalgae have been investigated. Such strategies include selecting the best performing species and strains and the optimization of culturing conditions, with special emphasis on the different cultivation systems and the effect of different abiotic factors on PUFAs’ accumulation in microalgae. Moreover, developments and results obtained through the most modern genetic and metabolic engineering techniques are described, focusing on the strategies that lead to an increased lipid production or an altered PUFAs’ profile. Additionally, we provide an overview of biotechnological applications of PUFAs derived from microalgae as safe and sustainable organisms, such as aquafeed and food ingredients, and of the main techniques (and their related issues) for PUFAs’ extraction and purification from microalgal biomass.

## 1. Introduction

Microalgae exploitability has received great attention for different purposes during the last decades. These microorganisms have been, indeed, largely investigated as a source of biofuels [1,2], bioactive components for cosmetic and cosmeceutical formulations [3,4], and high-value pharmaceuticals [5]. Microalgae appear also suitable as fertilizers [6] because of their water-binding capacity and their ability to release macro- and micronutrients to the surrounding soils at low rates [7]. They are currently being investigated for aquatic environments’ bioremediation, known as phycoremediation [8,9]. Furthermore, microalgae have been studied because of their potential for the production of healthy food and feed supplements [10,11]. Microalgal biomass can be used, indeed, as raw material for third-generation biorefinery processes [12], which foresee its conversion in a wide spectrum of marketable products and energies, minimizing costs and waste production. To date, various companies have adopted this approach and are able grow microalgae at pilot scale and to produce different kinds of commodities (mainly energy carriers, pigments, and polyunsaturated fatty acids) from the same biomass, for various biotechnological sectors [13,14,15].

Microalgae-based technology exhibits several advantages, which overcome numerous constraints related to the cultivation of other organisms and/or natural sources. They do not require arable land and, thus, do not compete with agricultural crops for space [16] and exhibit growth rates higher than those observed in terrestrial plants [17]. Besides, microalgal growth could be considered an environmentally safe process, not requiring herbicides and pesticides for maintenance. Their growth can be directly related to carbon dioxide fixation [18,19] and can also contribute to the reduction of other greenhouse gases in the atmosphere such as nitrogen and sulfur oxides [20]. Moreover, differently from lignocellulosic materials, microalgal biomass does not require cost- and time-consuming pre-treatments for the extraction of biomolecules [12] and lends itself to different extraction techniques, some of which minimize or avoid the use of solvents [21].

Polyunsaturated fatty acids (PUFAs) are fatty acids (FAs) containing two or more double bonds in their acyl chain. PUFAs are classified into two main groups on the basis of the length of their carbon backbone: short chain polyunsaturated fatty acids (SC-PUFAs), with 16 or 18 carbon atoms, and long-chain polyunsaturated fatty acids (LC-PUFAs) with more than 18 carbons (the Greek letter ω in PUFAs’ nomenclature indicates the first position of the double bond from the terminal carbon). It is common knowledge that PUFA consumption is related to various health benefits for humans, as well as the improvement of metabolic rates, the regulation of blood pressure and glucose level, and the protection against numerous diseases, including some types of cancer [22]. Clinical and epidemiological studies have shown that an eicosapentaenoic acid (EPA-C_20:5_ω3)-rich diet contributes to minimizing risks of cardiovascular diseases [23,24]. Docosahexaenoic acid (DHA-C_22:6_ω3) and arachidonic acid (AA-C_20:4_ω6) are important to avoid impairments in infant brain development and cognitive deficiency [25,26,27]. EPA and DHA may also lower the risks of obesity in both humans and animals [28] and can prevent chronic inflammatory diseases [29]. Humans, as other mammals, are unable or poorly able to synthesize some essential PUFAs, such as linoleic acid (LA-C_18:2_ω6) and α-linolenic acid (ALA-C_18:3_ω3), which are precursors of AA and DHA, respectively. Therefore, a direct uptake of these compounds from external sources is necessary. These sources include aquatic organisms (fish, molluscs, and crustaceans), animal-derived commodities (meat, milk, eggs), fungi, bacteria, and vegetable sources, such as some plants and microalgae [30]. The main microbial sources, which are able to produce highly valuable lipids for nutraceutical purposes, are summarised in Figure 1, while the main PUFAs, which may provide health benefits, are summarized in Table 1.

The use of vegetal sources of oils is also considered as a useful means to limit the total use of fishmeal in aquaculture. Due to the high commercial demand, and because of the competition from other sectors such as the pharmaceutical trade [31], fish-based commodities are, actually, very expensive. Moreover, aquaculture accounts for a large portion of the global demand for fish oil, leading to the paradox that fish oil is used for fish breeding. Aside from being uneconomic, fish oil production is also considered detrimental for the environment, since its increasing demand leads to a fast decline of natural fish stocks. Finally, vegetal alternatives to fish oil could significantly reduce risks of incorporating aquatic pollutants (carcinogen contaminants, heavy metals) and/or antibiotics used in fish farms and the occurrence of unpleasant odours and tastes.

Some microalgae-derived products are currently available on the market. Algal biochemical composition, and, subsequently, the yields of their high-value lipids can be tuned through regulation of growth conditions and/or genetic transformations aimed at modifying algal metabolism. Short-chain FAs with lower pureness and low unsaturation degrees, which are inadequate for the production of food or feed commodities and for pharmaceutical purposes, are only useful for biodiesel production. However, the high cost of microalgal biofuel compared to that of fossil sources still prevents its production. Conversely, different industrial plants producing food or feed supplements are spreading worldwide, even if there are still few species that have found a market. This review was aimed at describing the current knowledge of microalgae-derived oils with nutritional value, the strategies to improve their yields, and the main bottlenecks related to large-scale production by the food and feed industries.

## 2. Modulation of Growth Conditions to Enhance the Production of PUFAs

Lipid metabolism in microalgae can be controlled by modulating growth parameters, such as temperature, light, partial pressure of carbon dioxide (CO_2_), and nutrient supply. Optimization of such parameters can increase growth rates and, thus, biomass and lipid productivities. Changing the biochemical composition of microalgae by tuning physical and nutritional parameters leads to similar changes in the biochemical composition of microalgae-fed organisms, both at the larval and adult stages [32,33].

### 2.1. Temperature

Temperature can be considered a useful factor to select the most suitable strains for PUFA-rich polar lipid production [34]. Temperature shifts, indeed, play a key role in adjusting membrane fluidity: Generally, lower temperatures enhance the unsaturation degrees of FAs, and vice versa. By contrast, growth rates can decrease at decreasing temperature, leading to the decline in the productivity of all lipids including PUFAs; the temperature conditions yielding the greatest PUFA productivity are usually species specific.

The effect of low (≤20 °C) temperatures was evaluated on the oleaginous microalga *Scenedesmus obtusus* grown in outdoor plants. A two-fold increase in PUFAs with respect to indoor cultures, in which higher temperature values were constantly maintained, was detected [35]. This strategy was found to be effective also in diatom species: A drastic temperature decrease by 9 degrees (from 20 to 11 °C) stimulated PUFAs’ production in *Cylindrotheca closterium* during the stationary phase [36]. PUFAs were also found to reach high proportions (i.e., ca. 40%) of the total FAs in *Odontella aurita* cultured at 8 °C, compared with treatments at 16 °C and 24 °C, in which the proportion of PUFAs was 28–31% and 6–26%, respectively [37].

The concentration of PUFAs generally decreases at increasing temperatures. For example, EPA accumulation declined markedly with the increase of temperature also in the red alga *Porphirium purpureum* [38]. A significant decrease of EPA, DHA, and other PUFAs with increasing temperature (from 15 to 25 °C) was also observed in the diatom *Phaeodactylum tricornutum* [39].

### 2.2. Nutrients

Both culture media composition (e.g., nutrient concentration and composition) and growth phase (since nutrients tend to decrease over time during algal growth) can affect lipid metabolism. Generally, the concentration of unsaturated fatty acids (UFAs) is higher during the exponential phase and tends to decrease in aged cultures. This happens because plastidial membranes, which are actively biosynthesised during the exponential phase, contain larger proportions of UFAs compared to other cellular membranes as well as lipid droplets [40].

To our knowledge, nitrogen (N) limitation or depletion is the most common culturing manipulation used to enhance lipid production, although it usually causes a reduction in biomass and lipid yields [41,42,43]. Two-stage cultivation systems—the first one of nutrient replenishment, to increase biomass, and the second one with low or null N supply, to enhance lipid production—can be effective to overcome this problem [2,18] and are correlated to a strong increase of the neutral lipid fraction (especially triacylglycerols (TAGs)). For example, two-stage cultivation was found to be effective in increasing lipid concentration in *Nannochloropsis* spp. [44,45]. Since PUFAs in phospholipids (PLs) are nutritionally more available than those esterified in TAGs [46], an efficient production process might be achieved by maximising the proportion of PLs within cells and that of PUFAs within PLs. This can be usually obtained by culturing microalgae under N limitation rather than incubating microalgal biomass in N-depleted media. In *Isochrysis* aff. *galbana* (clone T-Iso) and in *Nannochloropsis oceanica*, the proportion of EPA in PLs was, indeed, lower under total depletion of N, and a similar trend was observed for DHA in T-Iso [47]. N limitation likely led to a decrease in the production rates of ω-3 PUFA, stimulating TAGs’ accumulation [39,48]. Similar results were obtained for *P. purpureum*, where EPA were mostly concentrated under N replenishment than in the stationary phase, when an active cell division, and, thus, membrane lipid accumulation were still in progress [49].

Phosphorus (P) limitation triggered an increase in the concentration of UFAs with respect to the total FAs, since P depletion stimulated the activity of Δ6-desaturase and, thus, conversion from palmitic acid (PA-C_16:0_) to LA acid. Moreover, stimulation of the ω-6 pathway and inhibition of the enzyme Δ-17 desaturase under low P concentration selectively promoted the biosynthesis of AA, to the detriment of the EPA pool [50]. Similar results were obtained with the freshwater eustigmatophyte *Monodus subterraneus*, where EPA and LC-PUFAs were drastically reduced under P (K_2_HPO_4_) limitation in a dose-dependent way [51].

Silicon (Si) limitation seems to affect negatively PUFAs’ production in diatoms [52] and is likely to be more effective for the accumulation of neutral lipids for biofuel production, since oils with little PUFA content are highly suitable for biofuel development [53].

The relative proportion of PUFAs over the total biomass as well as PUFA composition are dependent upon growth phase. For example, EPAs were found to be more abundant during the late stationary phase, while the DHA-to-EPA ratio (which must be comprised between 1 and 2 for larval fish feed) decreased since no increase of DHA levels in *P. tricornutum*, *Thalassiosira weissflogii*, *Thalassiosira pseudonana*, and *Rhodomonas salina* was observed [54].

### 2.3. Salinity

Salinity stress causes species-specific reactions in different microalgal strains. It causes variations in the biochemical composition of microalgae, leading to a decrease in the energy available both for photosynthesis and for intracellular FAs’ composition changings, as a protection against osmotic stress [55]. It has been studied both in freshwater and saltwater species and generally causes changes in UFAs’ concentrations and typology in algal strains when salt concentration is above or below the optimal level. High salinity stress (ca. 40), for example, triggered lipid and PUFAs’ production in the green microalga *Tetraselmis* sp., which reached a high (16%) percentage of PUFAs under optimal conditions of light intensity and pH [56]. While the proportion of lipids over the biomass tends to increase at increasing salinities, PUFA content typically decreases in *Nannochloropsis* spp. For example, *N. oceanica* UTEX2379 was found to increase its content in total FAs at increasing salinities (13 to 40 g L^−1^ NaCl) while EPA content peaked at 13 g L^−1^ NaCl [44]. Similarly, Martínez-Roldán et al. [57] observed a sharp increase in total lipids at 54 and 81 g L^−1^ NaCl for *N. oceanica* CCALA804 and *N. oceanica* CCMP1779, which exhibited the highest content of saturated FAs when cultured at a salinity of 50 compared to 10 and 35 [58].

The freshwater green alga *Golenkinia brevispicula* was grown under a two-stage cultivation system, with a first phase—semicontinuous batch production in a freshwater medium—aimed at increasing biomass, and a second one—batch conditions under high salinity—in which 35 g L^−1^ marine salts were added to the culture in order to increase both carotenoid and lipid content within microalgal biomass. Under salinity stress, PUFAs’ percentage was almost halved with respect to the previous phase of nutrient replenishment, and it was likely due to a stronger regulation of membrane structure and fluidity in the semicontinuous phase, during which an active cell division was observed [59]. Conversely, the addition of 0.2% NaCl in cultures of *Trachydiscus minutus* led to a slight increase of EPA content, but a further salinity increase (0.8%) totally inhibited growth [60]. *Nannochloropsis* spp. were found to increase their lipid content when cultured at salinities higher than seawater values [45,57], and the percentage of PUFAs over the total FAs typically decreases at increasing salinities [44].

Similar results were observed for the green alga *Chlamydomonas reinhardtii* (strain 137C), where NaCl-induced stress determined an accumulation of saturated FAs, to the detriment of unsaturated ones, and an up-regulation of genes involved in fatty acid biosynthesis (prolyl hydroxylase domain 2-PDH2, Acetyl-CoA Carboxylase-ACCase, MAT and 3-ketoacyl-ACP-synthase-KAS2) [61].

### 2.4. Irradiance

Light intensity can affect lipid metabolism as a result of membrane biosynthesis or storage lipid accumulation in case of excessive carbon accumulation. While saturated FAs tend to increase at high light irradiance, the relative proportion of PUFAs over the dry weight decreases or does not change significantly. For example, very high irradiance (750 µmol photons m^−2^ s^−1^) increased saturated FAs in *P. tricornutum* with respect to lower light intensities (150 µmol photons m^−2^ s^−1^); however, changes in the expression of genes responsible for PUFA production did not elucidate the mechanisms of their modulation [62]. These results were consistent with those obtained by Qiao et al. [39], which evaluated the effect of lower light intensities (50–150 µmol photons m^−2^ s^−1^) on *P. tricornutum* and found that an irradiance level of 150 µmol photons m^−2^ s^−1^ stimulated DHA production. Similarly, *N. oceanica* UTEX2379 increased its content in total FAs and did not exhibit significant changes in EPA when cultured at 700 µmol photons m^−2^ s^−1^, compared to 150 µmol photons m^−2^ s^−1^ [44]. *N. oceanica* CCMP1779 also contained far greater amounts of saturated FAs at 300 µmol photons m^−2^ s^−1^, compared to 25 µmol photons m^−2^ s^−1^ [58].

### 2.5. Autotrophic, Heterotrophic, and Mixotrophic Conditions

Several microalgal genera, including *Phaeodactylum* and *Scenedesmus*, are obligate autotrophs, but others are able to grow in the presence of organic substrate as facultative (some *Nannochloropsis*, *Dunaliella,* and *Tetraselmis* species) or obligate (the dinoflagellates *Crypthecodinium* and *Gyrodinium*) heterotrophs [63,64]. Among heterotrophs, also Thraustochytrids, a fungus-like class of Stramenopiles, are considered a suitable source of DHA, which can accumulate at low temperatures and salinities [65]. Some algal species, such as *Galdieria sulphuraria*, showed a higher PUFA content when cultured under heterotrophic rather than phototrophic conditions, where a higher content in saturated and monounsaturated FAs was observed [66]. Similar results were obtained with the marine diatom *Cyclotella cryptica*, in which heterotrophic conditions (i.e., culture media enriched with glucose) were found to be effective to increase the EPA yields with respect to the autotrophic regime [67]. Production of PUFAs under photoheterotrophic conditions can be also influenced by the carbon source: Under controlled (indoor) conditions, for example, the employment of glucose was found more effective than other organic substrates (fructose, sucrose, and maltose) to stimulate PUFAs’ production in *Tetraselmis gracilis* and *Platymonas convolutae* [68]. Unfortunately, glucose is also one of the more expensive carbon sources, and this can render the production of highly valuable lipids uneconomic [69,70,71].

Microalgae have also the ability to utilize nutrients from wastewater; this can minimize costs and water requirement and contribute to bioremediation of effluents [70]. To the best of our knowledge, only a few papers regarding highly valuable lipids for nutritional purposes obtained by waste-fed microalgae are available in literature. However, their efficiency depends on the strain and on waste typology. The employment of digestate (e.g., biodegradable feedstock from anaerobic digestion) as nourishment for the green alga *Chlorella vulgaris*, for example, was detrimental for the total lipid pool and its unsaturated fraction [71]. Conversely, a partial (25%) replacement of the synthetic medium with industrial process waters was not inhibitory for the growth of the freshwater species *Arthrospira platensis* and stimulated PUFAs’ (42–45% of total lipid pool) accumulation [72]. Recent works demonstrate that alternative organic sources such as glycerol wastes can be successfully used to sustain microalgal growth of heterotrophic species (e.g., *Schizochytrium* sp.), improving the yields of DHA with economic and environmentally sustainable processes [73,74]. Biomass and DHA yields could be further optimized by nutrient supply [73] and by an appropriate tuning of both physical parameters and organic substrate concentrations [74]. These results are particularly interesting since glycerol is the main by-product of biofuel production; so, they pave the way for an eligible biorefinery approach, which couples the production of metabolites for the food and feed industries with the production of green energy in the same route.

### 2.6. CO_2_

CO_2_ is the main carbon source for autotrophic organisms, including microalgae, and can stimulate or inhibit growth, depending on its concentration and microalgal tolerance. Adequate CO_2_ levels can optimize photosynthesis by enhancing its concentration in proximity of the enzyme ribulose-1,5-bisphosphate carboxylase-oxygenase (RuBisCO), and, in turn, algal growth and productivities. High CO_2_ supply is often counterbalanced by a reduction in oxygen (O_2_) concentration. Previous works have shown that low O_2_ levels affected enzymatic desaturation in cyanobacteria [75], increasing PUFAs within their biomass. A similar trend was observed in the freshwater species *Scenedesmus obliquus* and *Chlorella pyrenoidosa*, where PUFAs increased at increasing CO_2_ concentrations [76]. A slight increase of unsaturated acids occurred also in *Nannochloropsis* sp. (strain MASCC11) in cells exposed to high (5–15%) CO_2_ percentages [77]. The marine oleaginous microalga *Microchloropsis gaditana* showed both higher productivities and an increase of LC-PUFAs when exposed to high (3% *v/v*) carbon dioxide levels with respect to air-insufflated (e.g., only 0.03% *v/v* CO_2_) cultures [78].

## 3. Genetic Engineering for PUFAs Production

In the last years, genetic and metabolic engineering have been used to improve different features of organisms, for example, increasing the production of high-added-value biomolecules.

High-throughput sequencing technologies and genetic transformation techniques greatly contributed to enhance lipid production, and, in particular, PUFAs’ accumulation, not only in microalgae but also in plants, yeasts, and bacteria. A number of strategies can be employed to regulate genes involved in lipid metabolism and PUFAs’ biosynthesis, generally perturbing gene expression by overexpressing or silencing endogenous genes; common strategies are the overexpression of enzymes of the FAs’ or TAGs’ biosynthesis pathway, the perturbation of the regulation of related biosynthetic pathways, the block of competing pathways, and, in some cases, the use of a multi-gene transgenic approach [79,80]. Furthermore, several researchers highlighted the possibility of heterologous protein expression in different organisms: This means expressing microalgae proteins in other microalgae or in other organisms or expressing proteins from other organisms in microalgae. These studies allowed, firstly, characterizing a large number of genes involved in different metabolic pathways and, secondly, identifying interesting targets for increasing PUFAs’ production.

### 3.1. Genetic Transformation and Gene Perturbation in Microalgae

Several transformation methods were recently developed and optimized, allowing the introduction of genetic material into the cell nucleus and, hence, the genetic manipulation of numerous microalgal species: among the most relevant ones, there are the model microalgae *C. reinhardtii*, *P. tricornutum*, and *Nannochloropsis* spp., all promising lipid-producing species. Nuclear transformation in microalgae is possible through methods ranging from nanoparticles’ bombardment [81] and agitation with glass beads [82], to the introduction of an episome via bacterial conjugation [83], *Agrobacterium* construction [84], and electroporation [85]. These techniques are mainly used to manipulate gene expression.

Different approaches ranging from random to targeted mutagenesis have been used in microalgae.

The random mutagenesis approach carried out throughout physical or chemical mutagens can be applied to species for which genetic transformations are not possible. This method causes alterations to the organism’s DNA, followed by selection of mutants with the desired metabolic properties [86].

Although random mutagenesis produced a lot of interesting results [86], the recent development of nuclear transformation and the availability of genome sequences for an increasing number of model species contributed to enlarge the collection of tools for targeted mutagenesis allowing overexpression and knock-down or knock-out of specific target genes in microalgae.

Gene overexpression is carried out by means of constructs containing strong regulatory sequences, driving an increased expression of the gene of interest. On the other hand, RNA interference (RNAi) is used for targeted gene downregulation (knock-down), which occurs through a variety of mechanisms, including translation, inhibition, RNA degradation, and/or transcriptional repression [87,88]. More recently, genome editing tools such as sequence-specific DNA nuclease technologies are playing an increasingly important and revolutionary role, enabling targeted modification of genomic sequences (knock-out) with high efficiency, and also in traditionally genetically intractable species. Among DNA nuclease technologies, TALENs (transcription activator-like effector nucleases) and CRISPR/Cas9 (clustered regulatory interspaced short palindromic repeats) are being used in microalgae. TALENs are chimeric proteins that contain two functional domains, a DNA-recognition transcription activator-like effector (TALE), and a DNA nuclease domain, which work together to recognize a specific DNA sequence and introduce a double-stranded break with an overhang [87,89], while the CRISPR/Cas9 system consists of a CRISPR-associated endonuclease (Cas protein), which cuts DNA in a specific target region determined by a short guide RNA, introducing a double-stranded break [87]. A variation of this last technique is CRISPR interference (CRISPRi), which provides a complementary approach to RNAi with the difference that CRISPRi regulates gene expression primarily on the transcriptional level, while RNAi controls genes on the mRNA level [90].

An overview of the studies that have produced successful results through microalgae mutagenesis and genetic manipulation, along with several unsuccessful attempts, are presented on Table 2. Each study contributed to reconstruct the metabolic pathways of interest for PUFAs’ production and allowed identifying more promising targets for subsequent biotechnological applications.


### 3.2. Enhancement of the Fatty Acid Biosynthetic Pathway

The main approach includes the direct modification of FAs’ composition through the manipulation of the genes involved in FA biosynthesis (Figure 2), generally by increasing the expression of one or more enzymes.

The malonyl-CoenzymeA (malonyl-CoA) generated by ACCase enters into de novo FA synthesis: Malonyl-CoA is first converted to malonyl-Acyl Carrier Protein (ACP) by Malonyl CoA-ACP transacylase (MCAT) [183] (Figure 2). Overexpression of the endogenous gene encoding MCAT in *N. oceanica* resulted in a 31% increase in neutral lipids along with changes in FAs’ composition with EPA increased by 8% [106]. A similar approach in the heterotrophic protist *Schizochytrium* sp. led to a 172.5% increase in EPA, 81.5% increase in DHA, and 69.2% increase in docosapentaenoic acid (DPA-C_22:5_ω3) content [107]. KAS and acyl-ACP thioesterase (FAT) catalyzed the initial condensing reaction in FA biosynthesis [104] (Figure 2, sections 1 and 2). The overexpression of these three genes (a specific MCAT called MCTK, KAS, and FAT) in *Haematococcus pluvialis* led to a 2-fold increase in EPA content and more than 4-fold increase for DHA content, in addition to an overall increase of 32% total FAs’ content [104]. KAS genes were combined with a thioesterase (TE) gene (FAT-A or FAT-B) to obtain a mutant that produced an increased amount of C_8–16_ FAs in the microalgae *Chlorella* or *Prototheca* spp. [184]. Heterologous co-expression of a polyketide synthase-like PUFA synthase system (FAS) (Figure 2, sections 1 and 2) from *Schizochytrium* sp. and a phosphopantetheinyl transferases (PPTase) from *Nostoc* in canola (*Brassica napus*) seeds showed an increase in de novo synthesis of DHA and EPA from malonyl-CoA without substantially altering plastidial FAs’ production [175]. Overexpression of a *P. tricornutum* TE did not alter the FAs’ composition of *P. tricornutum*, but enhanced total FAs’ content by 72% and EPA content by 10% [103]. Lastly, Ozaki [185] provided an efficient lipid production method through heterologous expression of *Nannochloropsis* sp. Acyl-ACP TE in *Escherichia coli.*

The overexpression of enzymes catalysing the last steps of a biosynthetic pathway is often preferred and can easily lead to increased biosynthesis of the final product of the pathway. This is due to the fact that pathways are often controlled by a single enzyme typically catalysing the final reaction and thus the metabolic pathway flux (and the concentration of the final product of the pathway) [186]. It is, thus, physiologically more common to change a metabolic flux and the production of the final metabolite in the pathway than varying the intermediary concentrations [187]. Nevertheless, in this case, the most promising results were obtained from the overexpression of upstream enzymes such as MCAT [107]. Furthermore, an interesting strategy consists of the overexpression of several enzymes involved in the same biosynthesis pathway, such as MCAT, KAS, and FAT, which together allowed doubling up to a quadrupling of the amount of EPA and DHA final products [104].

### 3.3. Altering Elongation and Desaturation

De novo produced saturated FAs, PA, and stearic acid (SA-C_18:0_) can undergo desaturation by plastid or endoplasmic reticulum (ER) desaturases (DES) and elongation by different elongases (ELO), producing FAs with different degrees of unsaturation and with different chain length (Figure 2, section 3). Therefore, another more precise strategy to alter FA composition is to interfere with desaturation and elongation processes by selectively acting on elongases and/or desaturases.

Stearoyl-ACP desaturase (SAD) is a Δ9-DES that catalyzes the conversion of SA into oleic acid (OA-C_18:1_ω9). The reduction of activity of this enzyme could, thus, potentially lead to the accumulation of SA. It has been reported that gene silencing of SAD enhances SA content in *C. reinhardtii* [140]. After that, desaturase Δ12-DES converts OA into LA. Overexpression of Δ12-DES in *N. oceanica* significantly altered composition of total lipids and of individual lipid classes, with a drastic increase in an 18:2 proportion in phosphatidylcholine and in TAG under nitrogen starvation [110]. Some LA was converted further toward LC-PUFA, resulting in a 75% increase in AA [110]. The enzyme ω-3 desaturase (ω-3-DES or Δ15-DES) catalyzed the conversion of LA to ALA. Genetic modification in *C. vulgaris* by introducing the disrupted gene Δ15-DES through homologous recombination led to higher production of PA with a reduction in OA, with no remarkable changes observed in ALA composition [141]. The exogenous Δ15-DES gene from yeast in *Schizochytrium* led to greater cell size and lower polar lipid content than the wild-type (WT) strains [159]. In addition, the introduction of Δ15-DES improved the ω-3/ω-6 ratio from 2.1 to 2.6 and converted 3% DPA to DHA [159]. Desaturase Δ5-DES is the key enzyme responsible for AA production, but it can also convert eicosatetraenoic acid (ETA-C_20:4_ω3) into EPA. When the Δ5-DES gene was overexpressed in *P. tricornutum*, FA composition was altered, with a 65% increase in TAG content and 58% in EPA content. Engineered cells showed a similar growth rate to the wild-type, thus keeping high biomass productivity [109]. Another example of increasing PUFA biosynthesis is the heterologous expression of a Δ5-DES from the thraustochytrid *Thraustochytrium aureum* in another thraustochytrids, *Aurantiochytrium limacinum*: The amount of EPA in the transgenic thraustochytrids increased 4.6-fold, while AA content showed a 13.2-fold increase [152]. Moreover, a recent study demonstrated that overexpression of endogenous FA elongase genes in *T. pseudonana* can increase levels of EPA and DHA up to 1.4- and 4.5-fold, respectively [108]. Subsequently, Δ5-elongase (Δ5-ELO) catalyzed AA conversion into docosatetraenoic acid (DTA, C_22:4_ω6). The same enzyme can also convert EPA into DPA. *P. tricornutum* does not naturally accumulate significant levels of DHA [188], but it has been reported that the heterologous expression of the Δ5-ELO from *Ostreococcus tauri* in *P. tricornutum* resulted in an 8-fold increase in DHA, and the co-expression of an acyl-CoA-dependent desaturase (OtΔ6-DES) and an elongase (OtΔ5-ELO) from *O. tauri* into *P. tricornutum* displayed a further increase in DHA levels [151]. Substitution of fish oil with vegetable oil and fish meal with plant seed meals in aquaculture feeds reduced the levels of valuable ω-3 LC-PUFAs such as EPA and DHA and lowered the nutritional value due to the presence of phytate. In order to solve this problem, Pudney and co-workers engineered *P. tricornutum*, accumulating high levels of EPA and DHA together with exogenous proteins: the fungal *Aspergillus niger* PhyA or the bacterial *E. coli* AppA phytases, in combination with a *O. tauri* Δ5-ELO, increasing, respectively, DHA content (10–12%) [166].

The modification of FAs’ desaturation and elongation pathways alters their proportions inside cells and, depending on the target genes, allows obtaining greater quantities of specific FAs, which can have different subsequent applications. These molecules can be produced by the microalgae or by other organisms, depending on the final purpose. In some cases, for the study of a particular enzyme, the purification of the final product or the biotechnological application of interest, other organisms, such as plants, bacteria, or yeasts, may be preferable to the use of microalgae.

In this context, key enzymes present in microalgae can be heterologously expressed in crops, increasing the levels of PUFAs such as EPA and DHA, which are particularly interesting from a nutritional point of view, improving the nutritional values of final products deriving from those plants, which, thus, find application as feed, food supplements, or nutraceuticals [174,189].

For example, the heterologous expression of a Δ9-ELO gene from the microalga *I. galbana* in *Arabidopsis thaliana* resulted in an *A. thaliana* mutant able to biosynthesise the LC-PUFAs C_20:2_ω6 and C_20:3_ω3 [163,164]. The co-expression of a Δ9-ELO gene from *I. galbana* and a Δ8-DES from *Euglena gracilis* into *A. thaliana* genome led to a mutant able to produce two other LC-FA, the C_20:3_ω6 and the C_20:4_ω3 FAs [171].

Another research using edible plants as receiver species showed that seed-specific expression in transgenic tobacco (*Nicotiana tabacum*) and linseed (*Linum usitatissimum*) of cDNAs encoding FAs’ desaturases and elongases from the diatom *P. tricornutum* and the moss *Physcomitrella patens*, absent from all agronomically important plants, resulted in the very high FAs’ accumulation (30%), in particular, ω-6-desaturated C_18_ FAs and C_20_ PUFAs including AA and EPA [174]. Furthermore, Petrie and co-workers engineered an artificial pathway that produced 26% EPA in plant leaf triacylglycerol using a newly identified Δ6-DES from the marine microalga *Micromonas pusilla*, which converted LA into γ-linolenic acid (GLA-C_18:3_ω6). The Δ6-DES from *M. pusilla* was also demonstrated to function as an acyl-CoA desaturase with a preference for ω-3 substrates [173].

The heterologous expression of microalgal genes in other organisms can be an advantage when these other organisms can be cultured more efficiently than microalgae. Bacteria and yeasts have been the pioneering hosts for recombinant protein production because of their fast growth and easy culturability. The genetic structure and physiology of bacteria and yeasts are well known, making metabolic pathway modification through genetic engineering easier than in other organisms. Furthermore, the heterologous expression in bacteria or yeasts can allow a more in-depth study of specific enzymes through biochemical and functional analyses, which are often more complicated in microalgae.

EPA production from *E. coli* is being considered as an alternative and economic source for industrial and pharmaceutical sectors. A novel Δ6-ELO, which converts GLA into dihomo-γ-linoleic acid (DHGLA-C_20:3_ω6), and a novel Δ5-DES were isolated from *Isocrysis* sp. and *Pavlova* sp., respectively, and expressed in bacteria, obtaining a strong increase in AA and EPA content [181,182].

As regards studies carried out on yeasts, to enhance EPA production in *Mortierella alpina* by favouring the ω-3 pathway, a plasmid harboring the Δ6-DES gene from the microalga *M. pusilla* was constructed and overexpressed in an uracil-auxotrophic strain of *M. alpina*, increasing EPA production by 26-fold [180]. Moreover, Δ5- and Δ6-DES from *P. tricornutum* were cloned and characterized in *Saccharomyces cerevisiae* by Domergue et al. [179]. Using both desaturases in combination with *P. patens* Δ6-ELO, AA and EPA biosynthetic pathways were reconstituted in yeast, achieving an increase in AA and EPA in mutants [179]. Heterologous expression of *P. tricornutum* Δ5-ELO in the yeast *Pichia pastoris* allowed characterizing this enzyme capable of elongating AA and EPA [176]. Co-expression of *P. tricornutum* Δ5-ELO and *Isochrysis sphaerica* Δ4-DES (a desaturase that catalyses DTA conversion into DPA) to assemble the high-efficiency biosynthetic pathway of DHA in the transgenic yeast significantly increased DHA content [176]. Lu and co-authors [177] expressed Δ2-DES from the psychrotrophic Antarctic *C. vulgaris* in *S. cerevisiae*: The accumulation of Δ2-DES induced an interesting accumulation of linoleic acid content in expressing yeast, which was not present in wild-type strain [177].

### 3.4. Enhancement of the TAG Biosynthetic Pathway

In microalgae, TAGs are mostly biosynthesized by the ER-localized pathway. An alternative biosynthetic mechanism of plastidial origin was proposed in some microalgae, such as *C. reinhardtii* and *P. tricornutum*, and the resulting TAGs were found to be stored in lipid droplets in both cytosol and plastids [91,190]. The saturation and length properties of the FAs’ acyl chain was determined by the substrate specificity of the acyltransferases of the de novo TAG biosynthetic pathway [91,191,192], so that altering it can improve PUFAs’ production [93].

Glycerol-sn-3-phosphate acyl-transferase (GPAT) catalyses the first reaction of TAG synthesis, esterifying the acyl-group from acyl-CoA to glycerol-3-phosphate (G3P), forming lysophosphatidate (Lyso-PTA) via the Kennedy pathway (Figure 2, sections 2 and 4). This enzyme was overexpressed in several microalgal species for its characterization and for its involvement in lipid metabolism. A GPAT was identified and functionally characterized in the diatom *P. tricornutum*. Its overexpression led to a significantly higher proportion of unsaturated FAs compared with the wild-type. EPA levels increased by 40%, while those of C_16:0_ and monounsaturated FAs (MUFAs) decreased by 45% and 12%, respectively [92]. Balamurugan et al. [91] identified AGPAT1 in *P. tricornutum*, localized on the plastid membrane in this species, whose overexpression significantly altered the primary metabolism, with increased total lipid content but decreased content of total carbohydrates and soluble proteins. *P. tricornutum* AGPAT1 overexpression coordinated also the expression of other key genes involved in TAG synthesis, increasing TAG content by 1.8-fold with a significant increase in PUFAs, particularly EPA and DHA by 1.5-fold. Additionally, Jiang et al. [193] provided a method for increasing the lipid content, in particular TAG content, of diatom through GPAT overexpression in *P. tricornutum*. Another example is the heterologous expression of a GPAT gene from *Lobosphaera incisa*, which could reach up to 50% TAG content, in *C. reinhardtii*, showing an increase in FA content by 1.5-fold [150].

Diacylglycerol acyl-transferase (DGAT) catalyzes the formation of TAG from diacylglycerol (DAG) and Acyl-CoA (Figure 2, sections 2 and 4). This reaction is the terminal step in TAG synthesis and it is essential for subsequent lipid droplets’ formation. Niu et al. [93] showed that the overexpression of endogenous DGAT2 in *P. tricornutum* resulted in a 35% increase in neutral lipid content and a 76% increase in EPA content; the latter increase was also observed by Dinamarca et al. [94], along with a doubling in TAG content. Moreover, Zou et al. [99] used the strong constitutive promoter Pt211 to drive the expression of multiple target genes, such as GPAT and DGAT2, in *P. tricornutum*, resulting in a 2.7-fold increase in total lipids compared to the WT and a significant increase in EPA content, reaching up to 57.5% of dry cell weight. Yi and Xy [194] overexpressed a DGAT gene in *N. oceanica* to increase the specific PUFA content in TAGs. A heterologous expression plasmid containing the DGAT gene from *C. reinhardtii* was transformed into *S. obliquus*: This was the first report of successful genetic manipulation of *S. obliquus* to increase both biomass (that increased to 29%) and lipid content (85%) in *S. obliquus* [195]. In addition, the heterologous expression of DGAT2 from the plant *Brassica napus* in *C. reinhardtii* led to a decrease of total FAs, but enhanced the PUFA content, especially ALA, with an increase by up to 12% [156]. In another study, DGAT2 from *S. cerevisiae* was expressed in *P. tricornutum* to enhance TAG accumulation as well as to modify the FA composition in TAGs, in favour of high EPA levels [157]: Although TAG levels were enhanced by 2.3-fold, FA composition remained unchanged [157]. A DGAT1 gene derived from *C. ellipsoidea* was transformed in yeast, plant cells, and microalgae, obtaining an increase of the total FA content in the cells [196].

All the studies described here showed that GPAT and DGAT play a key role in lipid metabolism in microalgae. Overexpressing genes encoding for these two enzymes leads to a general increase in lipid content, with a different composition depending on the species engineered.

### 3.5. Inhibition of Starch and Other Complex Polysaccharides’ Biosynthesis

A possible strategy to increase lipid accumulation consists of blocking competing pathways that lead to the accumulation of energy-rich storage compounds, such as starch and other complex polysaccharides, to enhance the metabolic flux channelling to lipid and FAs’ biosynthesis [161]. In fact, starch is a major carbon and energy storage compound in many microalgae. The inactivation of ADP-glucose pyrophosphorylase (AGPase) (Figure 2, section 1) or of isoamylase by X-ray mutagenesis and UV mutagenesis in a *Chlamydomonas* starchless mutant led to a 3.5-fold increase in total lipid content and, for AGPase silencing, also to a 10-fold increase in TAG content, suggesting that the photosynthetic carbon from starch is driven towards TAG synthesis [123,197]. In diatoms, the fixed carbon is stored in vacuoles as chrysolaminarin, a water-soluble polysaccharide whose synthesis shares common carbon precursors with lipid synthesis. The effects of UDP-glucose pyrophosphorylase (UGPase) (Figure 2, section 1) suppression on chrysolaminarin biosynthesis and carbon allocation in *P. tricornutum* were investigated in two studies: Daboussi et al. [124] reported a 45-fold increase in TAG accumulation in the transgenic strains, generated through the disruption of the UGPase gene using TALEN, while Zhu et al. [125] demonstrated that silencing of UGPase through antisense or inverted repeats resulted in significant decreases in chrysolaminarin content and increases in lipid synthesis. On the other hand, antisense knockdown of the chrysolaminarin synthase gene in *T. pseudonana* cells reduced the accumulated chrysolaminarin and led to a 3-fold increase in TAG level, with minimal detriment to growth [126].

### 3.6. Altering Pyruvate Metabolism

The conversion of phosphoenolpyruvate (PEP) to oxaloacetate is catalyzed by phosphoenolpyruvate carboxylase (PEPC), which also plays a key role in photosynthesis [79] (Figure 2, section 5). FAs’ biosynthesis requires PEP, which is successively converted to pyruvate, acetyl-CoA, malonyl-CoA, and then FAs. Previous studies showed that there was a negative correlation between the activities of PEPC, a key enzyme of the amino acid metabolic pathway, and lipid accumulation because PEPC and ACCase share a common substrate, pyruvate [198]. For this reason, the inhibition of the activity of PEPC may promote the flow of carbon from PEP to pyruvate and to Acetyl-CoA, generally to FAs synthesis, enhancing the oil content of cells [130]. Knockdown of PEPC1 in *C. reinhardtii* by the CRISPR/Cas9 system enhanced lipid content and lipid productivity by 74% and 94%, respectively [128]. Similarly, knockdown of PEPC1 in *C. reinhardtii* by RNAi decreased PEPC activity by 39–50% and increased TAG level by 20% [129]. Using a multi-gene approach, the expression of two *C. reinhardtii* PEPC genes (PEPC1 and PEPC2) was down-regulated by amiRNA-mediated knockdown technology, resulting in a strong increase in FAs’ content by 48% [130].

Pyruvate dehydrogenase (PDH), which catalyzes the conversion of pyruvate to acetyl-CoA, is under the control of pyruvate dehydrogenase kinase (PDK), which can phosphorylate and deactivate PDH (Figure 2, sections 1 and 5). PDH and PDK form the pyruvate dehydrogenase complex (PDC), an immediate primer for the initial reactions of de novo FAs’ synthesis (Figure 2, sections 1 and 5). Therefore, with the same purpose of promoting the metabolic flow from pyruvate towards the conversion to Acetyl-CoA and consequently to the synthesis of a greater quantity of FAs, it is possible to remove the PDH inhibitor. Due to antisense knockdown of PDK, neutral lipid content of transgenic *P. tricornutum* cells increased up to 82% but without any change in FAs’ content [132], while amiRNA- mediated knockdown of *C. reinhardtii* PDH led to a 50% decrease in FAs’ content [143].

### 3.7. Reducing Lipid Catabolism

Engineering efforts focused on increasing lipid biosynthesis or blocking the competing pathways, such as carbohydrate formation, have successfully increased the lipid content [199], but they often resulted in engineered strains growing at lower rates than WTs [122,161]. An alternative strategy to increase lipid accumulation consists of inhibiting lipid catabolism. Lipid breakdown plays a key role in quickly providing acyl groups for membrane reorganization as environmental conditions change, contributing to polar lipid synthesis during dark cycles and remobilizing cell membranes upon release from nutrient stress [133]. Targeted knockdown of lipid catabolism, and specifically lipases, could potentially improve the accumulation of FAs and enhance lipid content with less impact on primary carbon pathways associated with growth [200].

Trentacoste and co-workers, using antisense RNAi constructs, successfully engineered *T. pseudonana* targeting an enzyme with lipase activity (Figure 2, section 5): This strategy resulted in an increase in TAG accumulation and total lipid yield without impacting growth rate, under both continuous light and alternating light/dark, continuous growth, and nutrient-replete versus nutrient-deficient conditions. FAs’ analysis revealed that knockdown strains contained greater amounts of PA and LA than wild-type species, with a 4-fold increase in EPA and a 3.2-fold increase in DHA content [133]. In *P. tricornutum*, TGL1 (a putative TAG lipase) knockdown mutant strains were created using an antisense RNA approach, showing a strong increase of TAG and PUFA in the lipid extracts in the mutant cell lines [135]. Knockdown of an OmTGL lipase in *P. tricornutum* considerably enhanced neutral lipid content at the stationary growth phase, with TAGs increased by 1.4 folds and a 70% increase in EPA content in the OmTGL RNAi lines [134]. Another possibility is to block some of the β-oxidation steps, as done by Kong et al. [137] in *C. reinhardtii*. The importance of peroxisomal FA β-oxidation in algal physiology was shown by the impact of the insertional mutation on FA turnover during day/night cycles and by the accumulation of 20% more oil under nitrogen depletion [137].

### 3.8. Overexpression of Transcription Factors

Transcription factors (TFs) bind to *cis*-acting elements in target gene promoters, controlling and regulating gene expression (Figure 2, section 6). TFs can be used as building blocks and regulatory tools in metabolic engineering and synthetic biology [201]. As an example, they can be engineered in host cells to control the expression of key enzymes in biosynthetic gene clusters [202].

Overexpression of DOF-type transcription factor genes in *C. reinhardtii* resulted in an increase of total lipids by ca. 2-fold [112]. Heterologous expression of DOF4 from soybean (*Glycine max*) significantly increased the lipid content of *Chlorella ellipsoidea* by 53% without growth inhibition [167]; such increase was attributed to the large number of genes with up-regulated expression, especially the ACCase gene [88,167]. TFs with the basic leucine zipper (bZIP) domain have been known as stress regulators and are associated with lipid metabolism in plants. Overexpression of a bZIP TF in *N. salina* resulted in enhanced growth with concomitant increase in lipid contents and phenotypes, also notable under stress conditions including N limitation and high salt [116]. Transcriptional engineering is a promising technique to enhance microalgal lipid production, in particular, altering the expression of multiple genes involved in the same metabolic pathway simultaneously [203]. Overexpression of phosphorus stress response-1 (PSR1) in *C. reinhardtii* can lead to changes in carbon storage metabolism through the control of specific genes involved in lipid and starch metabolisms. In detail, Overexpression of PSR1 can increase TAG accumulation without inhibiting the growth [115] or increasing starch biosynthesis, limiting neutral lipid accumulation [114,204]. Moreover, heterologous expression of three subunits of the transcription factor NF-Y from *C. ellipsoidea* in the plant *A. thaliana*, enhanced biomass production of 44.9–51% and total FAs’ production of 11.2–15.4% [196,205,206]. Lastly, Yang et al. [207] overexpressed a bHLH2 gene, encoding a basic helix-loop-helix transcription factor, in *N. salina,* enhancing lipid productivity and growth rate of microalgae by culturing it in a nitrogen-depleted medium and/or an osmosis stress medium. On the other hand, research by Ajjawi and co-workers represents the only example of TF knockout, obtained with the CRISPR/Cas9 method, which has shown effects on lipid production. Knockout of the homolog of fungal Zn(ii)_2_Cys_6_ encoding gene in *N. gaditana* improved partitioning of total carbon to lipids from 20% (WT) to 40–55% (mutant) in nutrient-replete conditions, but with a strong growth reduction. In the same study, Zn(ii)_2_Cys_6_ knockdown through RNAi increased twice as much lipid than in the WT, with little effect on growth [147].

As a metabolite sensor and gene expression regulator, TFs play an important role in determining the end product productivity in a cell factory. Therefore, the importance of TFs’ engineering is that it is a critical tool in optimizing phenotypes, as they are key components used to construct synthetic genetic circuits in vivo. However, the molecular mechanism, compatibility, robustness, and interaction of TFs in regulating metabolic networks in microalgae need to be understood and thoroughly elucidated to enhance the efficiency of TF-based strain development [208].

## 4. Potential of Microalgae for Sustainable Aquaculture

The global aquaculture production has been growing at an ever-increasing speed over the last two decades. This trend has been mostly driven by the increase in the world population as well as the ‘nutrition transition’ from a calorie-rich cereal diet to protein-rich meat diets that mostly include beef, poultry, pig, sheep, and fish. In particular, global demand for fish has significantly increased, on the basis of their ω-3 FAs’ proven health benefits. Dietary ω-3 PUFAs play indeed crucial and physiologically complex roles in several key metabolic functions [209]. In this context, aquaculture is one of the strategies that can contribute to limit over-fishing of wild populations, although there are growing concerns about aquaculture feed sustainability [55].

The major issues of aquaculture feed production are the limited availability of feed ingredients and their booming prices, such that, to date, fish meal and fish oil have proved to be the best compromise between economic constraints and a healthy composition of the final product [210].

However, aquafeed includes ocean-derived ingredients extracted from ‘forage’ fishes such as small- and medium-sized pelagic fishes. Generally, one-sixth of global capture fisheries are rendered into fish meal and fish oil commodities despite 90% of these harvested fish being food-grade for human consumption [211]. This is one of the economic, environmental, and sustainability concerns that make the reliance on fish meal and oil to produce aquaculture feed a limiting factor [212].

Different alternatives have been proposed to supplement or replace fish meal in aquaculture feeds, such as fish processing waste and plant-based ingredients. The latter, however, still needs to be supplemented with essential amino acids like methionine and lysine [213] and can also contain anti-nutritional components that compromise digestion. In this context, microalgae would be a more suitable replacement, since they are a natural source of ω-3 FAs and have a balanced amino acid profile [214]. To date, a variety of microalgal genera, such as *Nannochloropsis*, *Phaeodactylum*, *Tetraselmis,* and *Isochrysis,* have been extensively used in the aquaculture industry [11,215,216]. A list of recent research based on the application of microalgae-derived PUFAs as aquafeed is reported in Table 3.

The heterotrophic eukaryotic genus *Schizochytrium* seems to be particularly suitable as aquaculture food, since it can contain up to 58% of crude lipid and high concentrations of DHA [258]. *Schizochytrium* sp. is, indeed, a high-quality candidate for complete substitution of fish oil in juvenile Nile tilapia feeds, providing an innovative means to formulate and optimize the composition of tilapia juvenile feed while simultaneously raising feed efficiency of its aquaculture and further developing environmentally and socially sustainable aquafeeds [220]. Several studies were carried out in order to completely replace fish meal and/or fish oil with *Schizochytrium* as feed for Atlantic salmon [209,219], rainbow trout [222], tambaqui [221], and red seabream [223], but also as partial replacement or additional inclusion of control diets [226].

The genus *Isochrysis* spp. has been extensively used in aquaculture industry for several decades [259] and makes one of the most promising genera for aquaculture because of its high ω-3 PUFAs content, which can reach up to 5.4% of dry biomass [234,260].

In some studies, *Isochrysis* spp. has been demonstrated to improve fish growth performance and lipid deposition and enhances total ω-3 FAs, DHA, and EPA contents in fish muscle and liver [233,234].

Another example is given by *Arthrospira* spp., cyanobacteria with substantial productivity (20–90 tons/ha/year) that have been cultured and used as food and feed supplements [261]. *Arthrospira* sp. was included in experimental diets of broiler chicken feed [239], as previously done for *Staurosira* sp. by Austic and co-workers [237], and, in diets for Nile tilapia, it improved feed utilization efficiency and enhanced the overall health status of Nile tilapia juveniles [238].

Because of the high nutritional value, the genus *Chlorella* has been used as a dietary protein source for marine and freshwater fishes to improve weight gain and carcass quality [240,243]. However, its use as aquafeed has been limited by the relatively high level of fibres, which would retard digestion and absorption of nutrients because of fish inability to excrete cellulase and efficiently use complex carbohydrates as energy sources [241,262].

*Nannochloropsis* spp. are known as a source of ω-3 FAs that can be cultured with high productivity (33.6–84.0 tons/ha/year) [11,261]. Archibeque et al. [263] and Gbadamosi and Lupatsch [248] reported that *Nannochloropsis* produced favourable nutritional and health performances similar to fish meal and better than soybean meal in the diet of rabbits and Nile tilapia. Different species of *Nannochloropsis* were demonstrated to be successfully included in diets for crustaceans and fishes [243,245] combined with other microalgae to create a blend that completely replaced fish meat [211,257] or constituted it alone an aquafeed [249].

Moreover, the green algal genus *Tetraselmis* is among the most important live food species in marine aquaculture. Using it as complete replacement or partial inclusion in diets resulted in improved growth performances, increased appetite, and immunity enhancement due to the bioactive components, such as ω-3 and ω-6 FAs [251,254].

It has been estimated that, within a few years, once microalgae production, harvesting, and biomass processing have been improved, the production costs of EPA and DHA from microalgae will be equivalent to that of fish oil [11].

## 5. Extraction, Purification, and Stability of PUFAs

To obtain the maximum PUFAs’ yield from microalgal, an efficient extraction method has to be combined with an effective purification technique. To date, a complete process for PUFAs’ isolation from microalgae has not been successfully established; so, the methods for extracting PUFAs from fish are currently being adapted [264].

Cell disruption is the first step of lipid extraction process. Several methods are suitable for cell disruption: bead milling, high-pressure homogenization, freeze drying, hydrodynamic cavitation [265], ultrasonication [266], microwave [267], pulsed electronic field [268], acid/ionic liquid/surfactant/algicidal treatments, osmotic chock [269], or hydrolytic enzymes [270,271]. Among these, the microwave oven method has been identified as the easiest and most effective in microalgae such as *Botryococcus* sp., *Chlorella vulgaris*, and *Scenedesmus* sp. [264,267]. According to Ryckebosch et al. [272], cell disruption might be not essential in some microalgal species, and the wet biomass can be directly suspended in the extraction solvent, where cell wall is naturally degraded. Therefore, physico-chemical properties of specific microalgae have to be taken into account to establish if cell disruption is a necessary step or can be avoided.

Subsequently, lipids can be extracted with different solvent methods. The basic principle is the separation of the homogenate in different phases, which can be isolated. Specifically, an organic solvent penetrates through the cell membrane and interacts with the lipids forming solvent–lipids complexes, which diffuse across the cell membrane and across the static organic solvent into the bulk organic solvent [273]. In this context, the solvent composition is a key factor influencing solvent extraction. The most popular lipid extraction protocol was assessed more than 60 years ago by Folch and co-workers [274]. This method was initially settled for the extraction and subsequent purification of lipids from animal tissues, but was found suitable for other kinds of samples, including wet and dry microalgal biomass. Unfortunately, this technique is based on the use of a 2:1 chloroform: methanol solvent mixture, so it foresees a high amount of toxic solvents. A fast lipid recovery can be also obtained with methyl-tert-butyl ether (MTBE)/methanol, which avoids the use of a toxic solvent such as chloroform, allowing comparable or better lipid yields of the Folch method [275]. Recently, Li et al. [264] reviewed all the methods applied for EPA and DHA from microalgae, concluding that chloroform/methanol/water, hexane/96% ethanol, 96% butanol/96% ethanol/water, and *n*-hexane/isopropanol were found to be effective for lipid extraction.

Solvent extraction techniques are characterised by high yields of lipids obtained from microalgae, but often require more subsequent purification steps to eliminate as much solvent as possible from the final product. The use of solvents can be, indeed, unsuitable to extract products for pharmaceutical and food industrial sectors. Recently, new solvent-free lipid extraction methods have been developed, such as supercritical fluid extraction (SFE). This technique uses supercritical fluids—of which the most used is CO_2_—which are a composite form of gas and liquid properties, existing above a critical temperature and pressure [276]. Supercritical CO_2_ is chemically inert, safe, low cost, nontoxic, and it is environmentally feasible with no access to organic solvents, allowing a greater purity of the final product [277]. Moreover, this process overcomes one of the biggest disadvantages of other methods, which is the degradation of extracts, by providing a non-oxidizing environment, and the low critical temperature (around 31 °C) also prevents the thermal degradation of extract [278].

Many PUFAs are widely applied in the health and pharmaceutical fields, so they need high purity levels, which can be reached through purification and/or enrichment techniques. Depending upon carbon chain length and degree of unsaturation, different methods are applied to enrich and purify of PUFAs [273]. They include winterization, molecular distillation, enzymatic purification, low-temperature crystallization, purification by urea inclusion, chromatographic separation, and supercritical fluid fractioning [264]. These different methods can be often combined to get the higher yield and extraction efficiency of PUFAs from microalgae. For example, Mendes and co-workers [279] developed a procedure to concentrate DHA from *C. cohnii* involving saponification and methylation in wet biomass for further winterization and urea complexation, reaching a 99.2% yield of total FAs.

Purity and composition of the microalgal oils are important, but these should also be stable during storage. In fact, PUFAs are easily oxidized, leading to off-flavors, a decrease of the nutritional value, and even formation of potentially toxic compounds [280]. Moreover, PUFAs are susceptible to chemical modifications, such as metal-catalyzed autoxidation and hydrogenation, because of their multiple carbon-carbon double bonds vulnerable to electrophilic attack [24]. Chen and co-authors [281] showed that storage of microalgal biomass leads to lipid hydrolysis, in particular when stored as a wet paste at temperatures above the freezing point, but sometimes also as spray- and freeze-dried microalgal biomass [272]. Ryckebosch et al. [280] investigated both the primary and secondary oxidation, highlighting that microalgae oils were more oxidatively stable than fish oils, probably due to the presence of more polar antioxidants such as polyphenols in microalgae. The authors also stated that the oxidative stability of the microalgae oils was also shown to be dependent on the solvent used to extract lipids. However, little is known about presence of hydrolytic enzymes in the extracts, so further research on oxidation of microalgae extracts is definitely needed.

## 6. Summary, Conclusions, and Perspectives

The main purpose of this review article was to highlight the potential of microalgae-derived lipids as sustainable sources or food and feed. Modulation of nutrient supply (typology and/or concentrations in culture media) and of physical parameters (temperature, light, irradiance) can enhance the yields of specific lipids; culturing conditions that maximise production and/or productivity of highly valuable lipids can be species-specific, and, in some cases, specific growth strategies (such as a two-stage cultivation system) are required to maximise both biomass and lipid content.

Genetic and metabolic engineering were other useful tools to enhance lipid production (especially unsaturated ones) and include different strategies. Among them, improving some steps of the lipid biosynthetic pathway, turning off competitive metabolic pathways, or decreasing lipid catabolism are the most investigated strategies to increase lipid production, modifying their quantities as well as their composition. All previously discussed studies reported encouraging results, acting on different key enzymes and microalgal species, allowing the identification of promising target genes and organisms for subsequent research. These studies allowed us also to better understand the metabolic pathways that regulate lipid metabolism, for example, shedding light on enzymes not yet functionally characterized. Another winning strategy but still being explored is the manipulation of transcription factors, able to regulate different metabolic pathways at the same time and therefore able to act on multiple levels.

Currently, large-scale production of microalgae-derived lipidic commodities for food and feed industries is mostly based on cultivation in open ponds, since they represent the most economical alternative for massive growth. On the other hand, closed photobioreactors should be intended for final products requiring high quality levels and very low or null concentrations of contaminants, such as nutraceuticals and pharmaceuticals, but their rigid condition control is still very expensive. An interesting strategy to alleviate them is to first extract their lipids for the production of biodiesel, due to their high lipid contents, and then process the remainder lipid-free material as protein- and fatty acid-rich products [282,283]. Finally, we believe that a further effort to decrease operational costs through sustainable processes and nutrient recycle is required by phycological research.

An additional advantage of indoor photobioreactors compared to outdoor open ponds lies on the strict regulations on genetically modified organisms, including microalgae. Genetic engineering of microalgae contributed to improve specific biochemical features of some species, and culturing the resulting mutants rather than the corresponding wild types proved to be more economically viable. Since the accidental introduction of genetically modified microalgae in the environments can lead to dramatic consequences, comparable to those caused by some allochtonous species, legislation does not allow outdoor cultivation of such organisms in most countries. In contrast, genetically modified microalgae could be safely cultured in indoor systems, guaranteeing an aseptic separation between the inner and the outer environment is provided.

In this context, microalgal production of PUFAs has great development opportunities, but at the same time needs to face some key challenges: technical breakthroughs, access to venture capital and regulatory, academic and industrial training with co-operation between them, and reduction of biomass production cost. Although commercial-scale cultivation of microalgae is quickly improving, production costs of microalgal biomass is yet far from competitive. The main reasons behind the large production costs are related to the little proportions of PUFAs within microalgal biomass, which may depend on the cell growth and on the extraction methods.

In conclusion, PUFAs are well known to contribute to human health and well-being. These molecules, with different applications such as food and feed, are often obtained from fish which, due to the constant increase in demand, is less and less a sustainable source. Microalgae have been identified as currently the most sustainable and environmentally safe source to be exploited for the production of PUFAs for human consumption, as well as for feeding terrestrial and aquatic animals. However, the world has not yet fully exploited this potential, as other sources of PUFAs are still more economically advantageous, with already optimized production and extraction methods and high yields. Therefore, research in microalgal biotechnologies should focus on searching for novel strategies to minimize production costs and to increase the lipid yields, in the coming decade. Different aspects can be addressed to make microalgal production viable at an industrial scale, such as optimization of growth conditions, upstream processing to increase PUFAs productivities and selecting between different downstream processing to decrease production costs.

## Figures and Tables

**Figure 1 molecules-26-07697-f001:**
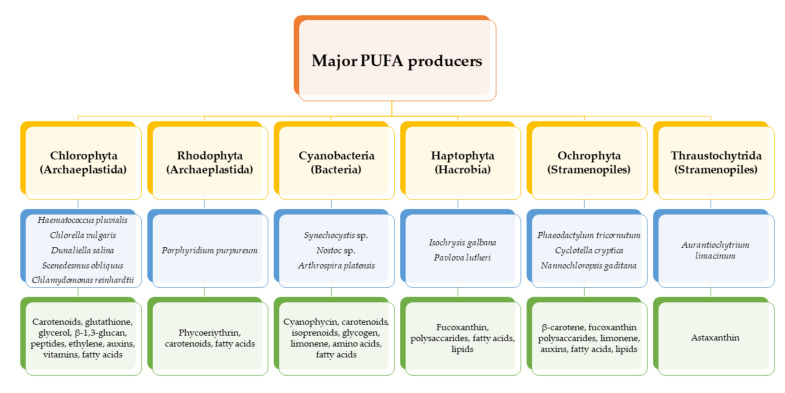
Main polyunsaturated fatty acid (PUFA)-producing microorganisms and associated high-value products.

**Figure 2 molecules-26-07697-f002:**
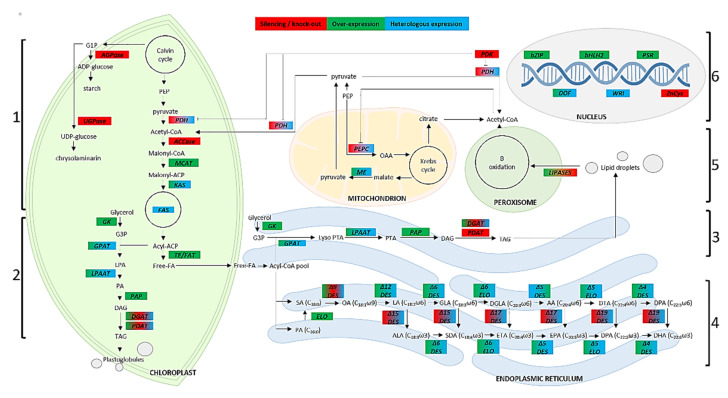
Schematic representation of microalgal lipid pathways. The enzymes are shown in squares with different colors, depending on the genetic manipulation that occurred: red for silencing or knock-out, green for overexpression, and blue for heterologous expression. Cellular organelles involved are plastid, endoplasmic reticulum, peroxisome, mitochondria cytosol, and nucleus. The numbers (from 1 to 6) near the square brackets indicate different figure sections. AA: Arachidonic Acid; ACCase: Acetyl-CoA Carboxylase; ACP: Acyl Carrier Protein; ADP-glucose: Adenosine-Diphosphate Glucose; AGPase: ADP-Glucose Pyrophosphorylase; bHLH2: Basic Helix-Loop-Helix transcription factor 2; bZIP: Basic Leucine Zipper transcription factor; CoA: Coenzyme A; DAG: Diacylglycerol; DES: Desaturase (the number near the Greek letter Δ indicates that the double bond is created at a fixed position from the carboxyl end of a fatty acid chain); DGAT: Diacylglycerol Acyltransferase; DGLA: Diacylglycerol Lipase Alpha; DHA: Docosahexaenoic Acid; DOF: DNA binding with One Finger -type transcription factors; DPA: Docosapentaenoic Acid; DTA: Docosatetraenoic Acid; ELO: Elongase; EPA: Eicosapentaenoic Acid; ETA: Eicosatetraenoic Acid; FAS: Fatty Acid Synthase; G1P: Glycerol-1-Phosphate; G3P: Glycerol-3-Phosphate; GK: Glycerol Kinase; GLA: γ-Linolenic Acid; GPAT: Glycerol-3-Phosphate Acyltransferase; KAS: Beta-Ketoacyl-Acyl-carrier-protein Synthase; LA: Linoleic Acid; LPA: Lysophosphatidic Acid; LPAAT: Lysophosphatidic Acid Acyltransferase; MCAT: Malonyl CoA-Acyl carrier protein Transacylase; ME: Malic Enzyme; OA: Oleic Acid; OAA: Oxalacetic Acid; PA: Palmitic Acid; PAP: Phosphatidic Acid Phosphatase; PEP: Phosphoenolpyruvate; PEPC: Phosphoenolpyruvate Carboxylase; PDAT: Phospholipid Diacylglycerol Acyltransferase; PDH: Pyruvate Dehydrogenase; PDK: Pyruvate Dehydrogenase Kinase; PSR: Phosphorus Stress Response transcription factor; PTA: Phosphatidic acid; SA: Stearic acid; SDA: Stearidonic Acid; TAG: Triacylglycerol; TE/FAT: Thioesterase/Acyl-ACP Thioesterase; UDP-glucose: Uracil-Diphosphate Glucose; UGPase: UDP-Glucose Pyrophosphorylase; WRI: WRINKLED1 transcription factor; ZnCys: Zinc/Cysteine transcription factor.

**Table 1 molecules-26-07697-t001:** Main polyunsaturated fatty acids (PUFAs): common and IUPAC names, acronyms, and structural formulas.

Fatty Acid	Acronym	IUPAC Name	Structure
Palmitic acid (C_16:0_)	PA	Hexadecanoic acid	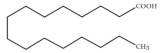
Stearic acid (C_18:0_)	SA	Octadecanoic acid	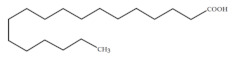
Oleic acid (C_18:1_ω9)	OA	(9Z)-Octadec-9-enoic acid	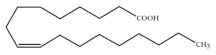
Linoleic acid (C_18:2_ω6)	LA	(9Z,12Z)-Octadeca-9,12-dienoic acid	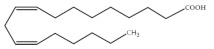
α-linolenic acid (C_18:3_ω3)	ALA	(9Z,12Z,15Z)-Octadeca-9,12,15-trienoic acid	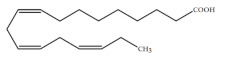
γ-linolenic acid (C_18:3_ω6)	GLA	(6Z,9Z,12Z)-Octadeca-6,9,12-trienoic acid	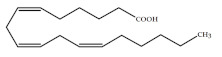
Eicosatetraenoic acid (C_20:4_ω3)	ETA	(8Z,11Z,14Z,17Z)-Icosa-8,11,14,17- tetraenoic acid	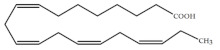
Dihomo-γ-linoleic acid (C_20:3_ω6)	DHGLA	(8Z,11Z,14Z)-Icosa-8,11,14-trienoic acid	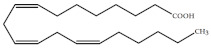
Arachidonic Acid (C_20:4_ω6)	AA	(5Z,8Z,11Z,14Z)-Icosa-5,8,11,14-tetraenoic acid	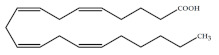
Eicosapentaenoic acid (C_20:5_ω3)	EPA	(5Z,8Z,11Z,14Z,17Z)-Icosa-5,8,11,14,17-pentaenoic acid	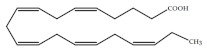
Docosatetraenoic acid (C_22:4_ω6)	DTA	(7Z,10Z,13Z,16Z)-Docosa-7,10,13,16-tetraenoic acid	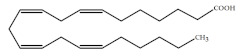
Docosahexaenoic acid (C_22:6_ω3)	DHA	(4Z,7Z,10Z,13Z,16Z,19Z)-Docosa-4,7,10,13,16,19-hexaenoic acid	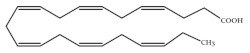
Docosapentaenoic acid (C_22:5_ω3)	DPA	(7Z,10Z,13Z,16Z,19Z)-Docosa-7,10,13,16,19-pentaenoic acid	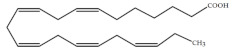

**Table 2 molecules-26-07697-t002:** Effect of different genetic modifications for enhancing PUFAs’ accumulation in microalgae. Studies are grouped by genetic modification, such as overexpression, silencing/knock-out, and heterologous expression, and then by the pathway in which the gene of interest is involved. The table represents the results obtained through specific genetic modifications. Genes and pathways are reported in Uniprot (https://www.uniprot.org/uniprot, accessed on 29 November 2021). Differences are shown as percentage increases, indicating the surplus over the wild-type control condition.

**Overexpression**					
**Gene**	**Pathway**	**Microalgal Species**	**Results**		**Reference**
AGPAT1	TAG biosynthesis	*P. tricornutum*	+50% EPA, +50% DHA content, +80% TAG content		[91]
GPAT	TAG biosynthesis	*P. tricornutum*	+41% PUFA content		[92]
DGAT2	TAG biosynthesis	*P. tricornutum*	+76% EPA content, +35% neutral lipid content		[93]
DGAT2	TAG biosynthesis	*P. tricornutum*	+100% total lipid, +80% TAG content, +20% EPA content		[94]
DGAT2	TAG biosynthesis	*N. oceanica*	+69% neutral lipid content		[95]
DGAT1A	TAG biosynthesis	*N. oceanica*	+39% TAG content		[96]
DGTT1-3	TAG biosynthesis	*C. reinhardtii*	no changes in lipid content		[97]
DGTT4	TAG biosynthesis	*C. reinhardtii*	+2800%TAG content		[98]
GPAT, DGAT2	TAG biosynthesis	*P. tricornutum*	+170% total lipid content		[99]
GK	TAG biosynthesis	*F. solaris*	+12% total lipid content		[100]
GPDH	TAG biosynthesis	*P. tricornutum*	+60% TAG content		[101]
LPAAT1	TAG biosynthesis	*C. reinhardtii*	+20% TAG content		[102]
thioesterase	FA biosynthesis	*P. tricornutum*	+72% total lipid, +10% EPA content		[103]
TE	FA biosynthesis	*P. tricornutum*	+16% EPA content		[103]
ACP, KAS, FAT	FA biosynthesis	*H. pluvialis*	+100% EPA, +340%DHA content, +32% total FA content		[104]
ME	FA biosynthesis	*P. tricornutum*	+150% total lipid content, −10% PUFA content		[105]
MCAT	FA biosynthesis	*N. oceanica*	+31% neutral lipid, +8% EPA content		[106]
MCAT	FA biosynthesis	*Schizochytrium* sp.	+172.5% EPA, +81.5% DHA, +69.2% DPA content		[107]
FA elongase	FA biosynthesis	*T. pseudonana*	+40% EPA, +350% DHA content		[108]
Δ5 desaturase	FA biosynthesis	*P. tricornutum*	+65% TAG content, +58% EPA content		[109]
Δ12 desaturase	FA biosynthesis	*N. oceanica*	+75% AA content		[110]
bHLH2	transcription factor	*N. salina*	+33% total lipid content		[111]
DOF	transcription factor	*C. reinhardtii*	+100% total lipid content		[112]
DOF	transcription factor	*C. reinhardtii*	+170% total lipid content		[113]
PSR1	transcription factor	*C. reinhardtii*	no quantified reduction of neutral lipid content		[114]
PSR1	transcription factor	*C. reinhardtii*	+10% TAG content		[115]
bZIP	transcription factor	*N. salina*	+50% total lipid content		[116]
PNPLA3	lipid turnover	*P. tricornutum*	+70% neutral lipid, +26% PUFA content		[117]
LDP1	lipid droplet metabolism	*P. tricornutum*	+30% total lipid, +40% neutral lipid content		[118]
ACCase	pyruvate metabolism	*C. cryptica*	no changes in lipid content		[119]
G6PD	carbohydrates metabolism	*P. tricornutum*	+170% total lipid content		[120]
NOA	nitric oxide metabolism	*P. tricornutum*	+80% neutral lipid, +400% TAG content		[121]
Silencing					
Gene	Pathway	Method	Microalgal Species	Results	Reference
AGPase	Carbohydrates’ metabolism	random mutagenesis	*C. reinhardtii*	+250% total lipids, +900% TAG content	[122]
isoamylase	Carbohydrates’ metabolism	random mutagenesis	*C. reinhardtii*	+450% total lipid content	[123]
UGPase	Carbohydrates’ metabolism	TALEN	*P. tricornutum*	+4400% TAG content	[124]
UGPase	Carbohydrates’ metabolism	RNAi	*P. tricornutum*	+4% total lipid content	[125]
CS	Carbohydrates’ metabolism	RNAi	*T. pseudonana*	+200% TAG content	[126]
SLM1	Carbohydrates’ metabolism	random mutagenesis	*S. obliquus*	+51% TAG content	[127]
PEPC1	pyruvate metabolism	CRISPRi	*C. reinhardtii*	+74% total lipid content	[128]
PEPC1	pyruvate metabolism	RNAi	*C. reinhardtii*	+20% TAG content	[129]
PEPC1, PEPC2	pyruvate metabolism	RNAi	*C. reinhardtii*	+48% FA content	[130]
PEPCK	pyruvate metabolism	RNAi	*P. tricornutum*	+40% total lipid content	[131]
CIS	pyruvate metabolism	RNAi	*C. reinhardtii*	+170% TAG content	[129]
PDK	pyruvate metabolism	RNAi	*P. tricornutum*	+82% neutral lipid, no changes in FA content	[132]
lipase	lipid turnover	RNAi	*T. pseudonana*	+300% EPA, +220% DHA content	[133]
omTGL	lipid turnover	RNAi	*P. tricornutum*	+70% EPA content	[134]
TGL1	lipid turnover	RNAi	*P. tricornutum*	+200% TAG, +10% EPA content	[135]
LIP1	lipid turnover	RNAi	*C. reinhardtii*	+150% TAG content	[136]
ACX2	Β oxidation	insertional mutagenesis	*C. reinhardtii*	+400% neutral lipid, +70% TAG content	[137]
MLDP	lipid droplet metabolism	RNAi	*C. reinhardtii*	no changes in TAG content	[138]
LDP1	lipid droplet metabolism	RNAi	*P. tricornutum*	−20% total lipid content	[130]
PDAT	TAG biosynthesis	RNAi	*C. reinhardtii*	general reduction of all TAG classes content	[139]
SAD	FA biosynthesis	RNAi	*C. reinhardtii*	+40% stearic acid content	[140]
ω-3-DES	FA biosynthesis	homologous recombination	*C. vulgaris*	no changes in PUFA and FA content	[141]
TES1	FA biosynthesis	TALEN	*P. tricornutum*	+70% TAG content	[142]
PDH	FA biosynthesis	RNAi	*C. reinhardtii*	−50% FA content	[143]
DGTT	FA biosynthesis	RNAi	*C. reinhardtii*	−35% TAG content	[144]
NR	N assimilation	TALEN	*P. tricornutum*	+20% TAG content	[145]
NR	N assimilation	RNAi	*P. tricornutum*	+43% total lipid content	[146]
ZnCys	transcription factor	RNAi	*N. gaditana*	+35% total lipid content	[147]
-	-	random mutagenesis	*P. lutheri*	+33% EPA, +33% DHA content	[148]
-	-	insertional mutagenesis	*N. oceanica*	+180% PUFA, +40% EPA content	[149]
**Heterologous expression**					
**Gene**	**Pathway**	**Source species**	**Receiver species**	**Results**	**Reference**
**Genes from microalgae in other microalgae**					
GPAT	TAG biosynthesis	*L. incisa*	*C. reinhardtii*	+50% FA content	[150]
ELO5	FA biosynthesis	*O. tauri*	*P. tricornutum*	+700% DHA content	[151]
ELO5, DES6	FA biosynthesis	*O. tauri*	*P. tricornutum*	+800% DHA content	[151]
Δ5DES	FA biosynthesis	*T. aureum*	*A. limacinum*	+360% EPA, +1220% AA content	[152]
ME	FA biosynthesis	*P. tricornutum*	*C. pyrenoidosa*	+220% neutral lipid content	[105]
(Bn)AccD, (Cr)ME	pyruvate metabolism	*B. napus, C. reinhardtii*	*D. salina*	+12% total lipid content	[153]
ACCase	pyruvate metabolism	*C. cryptica*	*N. saprophila*	no changes in lipid content	[119]
thioesterase	FA biosynthesis	*D. tertiolecta*	*C. reinhardtii*	+50% FA content	[154]
DGAT2	TAG biosynthesis	*C. reinhardtii*	*S. obliquus*	+85% total lipid content	[155]
**Genes from other organisms in microalgae**					
DGAT2	TAG biosynthesis	*B. napus* (plant)	*P. tricornutum*	+12% ALA content	[156]
DGA1	TAG biosynthesis	*S. cerevisiae* (yeast)	*P. tricornutum*	+130% TAG content	[157]
OLEO3	TAG biosynthesis	*A. thaliana* (plant)	*P. tricornutum*	+40% TAGcontent	[157]
(Sc)DGA1, (At)OLEO3	TAG biosynthesis	*S. cerevisiae* (yeast), *A. thaliana* (plant)	*P. tricornutum*	+260% TAG content	[157]
(Sc)G3PDH-GPAT-LPAAT, (Yl)DGATs	TAG biosynthesis	*S. cerevisiae, Y. lipolytica* (yeasts)	*C. minutissima*	+120% total lipid content	[158]
Δ3DES	FA biosynthesis	*S. dicilina* (yeast)	*Schizochytrium* sp.	+3% DHA content	[159]
ACP reductase	FA biosynthesis	*Synechocystis* sp. (cyanobacteria)	*C. merolae*	+133% TAG content	[160]
(Cc)C14-TE, (Uc)C12-TE	FA biosynthesis	*C. camphora, U. californica* (plants)	*P. tricornutum*	+80% TAG content	[161]
(Cc)C14-TE, (Uc)C12-TE, (Ch)KAS	FA biosynthesis	*C. camphora, U. californica, C. hookeriana* (plants)	*D. tertiolecta*	+4% FA content	[162]
C14-TE, C10-TE, ACP	FA biosynthesis	*C. lanceolata* (plant)	*C. reinhardtii*	general increase in different FAs classes content	[163]
(Bn)AccD, (Cr)ME	pyruvate metabolism	*B. napus* (plant), *C. reinhardtii*	*D. salina*	+12% total lipid content	[153]
ACC1	pyruvate metabolism	*S. cerevisiae* (yeast)	*S. quadricauda*	+60% FA content	[164]
ACC1, GDP1, GUT1	pyruvate metabolism	*S. cerevisiae* (yeast)	*S. quadricauda*	+50% total lipid content	[164]
ACS	pyruvate metabolism	*E. coli* (bacteria)	*Schizochytrium* sp.	no changes in lipid content	[165]
(An)PhyA, (Ot)Elo5	phytate metabolism	*A. niger* (yeast), *O. tauri*	*P. tricornutum*	+10% DHA, −25% EPA content	[166]
(Ec)AppA, (Ot)Elo5	phytate metabolism	*E. coli* (bacteria), *O. tauri*	*P. tricornutum*	+12% DHA, −18% EPA content	[166]
DOF4	transcription factor	*G. max* (plant)	*C. ellipsoidea*	+53% total lipid content	[167]
WRI1	transcription factor	*A. thaliana* (plant)	*N. salina*	+64% tota lipid content	[168]
**Genes from microalgae in other organisms**					
antisense PEPC	pyruvate metabolism	*Anabaena* sp.	*E. coli* (bacteria)	+47% lipid content	[169]
ACCase	pyruvate metabolism	*P. tricornutum*	*E. coli* (bacteria)	+100% neutral lipid content	[170]
Δ9-ELO	FA biosynthesis	*I. galbana*	*A. thaliana* (plant)	+18% PUFA content	[171]
Δ9-ELO (codon optimized)	FA biosynthesis	*I. galbana*	*A. thaliana* (plant)	+64% PUFA content	[172]
Δ6-DES	FA biosynthesis	*M. pusilla*	*A. thaliana* (plant)	+26% EPA content	[173]
(Ig)Δ9E, (Eg)Δ8D, (Ma)Δ5D	FA biosynthesis	*I. galbana, E. gracilis, M. alpina*	*A. thaliana* (plant)	+23% PUFA, +3% EPA, +7% AA content	[171]
(Pt)Δ5D, (Pt)Δ6D, (Pp)Δ6E	FA biosynthesis	*P. tricornutum, Physcomitrella patens*	*N. tabacum* (plant)	+30% PUFA content	[174]
(Pt)Δ5D, (Pt)Δ6D, (Pp)Δ6E	FA biosynthesis	*P. tricornutum, P. patens*	*L. usitatissimum* (plant)	+30% PUFA content	[174]
(Sc)PUFA-synthase, (No)PPTase	FA biosynthesis	*Schizochytrium* sp., *Nostoc* sp.	*A. thaliana* (plant) *B. napus* (plant)	+4% DHA, +1% EPA content	[175]
ELO5	FA biosynthesis	*P. tricornutum*	*P. pastoris* (yeast)	no quantified increase in DPA and DTA content	[176]
(Pt)ELO5, (Is)DES4	FA biosynthesis	*P. tricornutum, I. sphaerica*	*P. pastoris* (yeast)	+3% DPA, +2.35% DHA content	[176]
DES2	FA biosynthesis	*C. vulgaris*	*S. cerevisiae* (yeast)	no quantified reduction of LA content	[177]
DGTT2	FA biosynthesis	*C. reinhardtii*	*S. cerevisiae* (yeast)	+800% TAG content	[178]
(Pt)Δ5D, (Pt)Δ6D, (Pp)Δ6E	FA biosynthesis	*P. tricornutum, P. patens*	*S. cerevisiae* (yeast)	+0.23% EPA, +0.17% AA content	[179]
DES6	FA biosynthesis	*M. pusilla*	*M. alpina* (yeast)	+2500% EPA content	[180]
Δ6ELO	FA biosynthesis	*Isochrysis sp.*	*E. coli* (bacteria)	+6% SDA, +3% GLA content	[181]
(Iso)Δ6ELO, (Pav)Δ5DES	FA biosynthesis	*Isochrysis* sp., *Pavlova* sp.	*E. coli* (bacteria)	no quantified increase of AA and EPA content	[182]

**Table 3 molecules-26-07697-t003:** Microalgae and their derivatives used as feed in farming or aquaculture. The table includes microalgal species involved, application as feed, innovations, and nutritional values indicated as percentage increase respect to a basal diet.

Microalgal Species	Application (as Feed or Food)	Innovation	Nutritional Value (Respect to Basal Diet)	Reference
generic microalgae	Feed for light lamb *Ovis aries* farming	mixed with extruded linseed could, in part, replace fish meat	+520% ALA	[217]
generic microalgae	Feed for Pacific lamprey *Entosphenus tridentatus* aquaculture	complete replacement of yeast + fish oil	+21% total LC-PUFA; +9% total ω3 PUFA; +11% EPA	[218]
*Schizochytrium* sp.	Feed for Atlantic salmon *Salmo salar* aquaculture	complete replacement of fish oil	+100% DHA	[219]
*Schizochytrium* sp.	Feed for Nile tilapia *Oreochromis niloticus* aquaculture	complete replacement of fish meat	+23% total PUFA; +30.2% DHA	[220]
*Schizochytrium* sp.	Feed for tambaqui *Colossoma macropomum* aquaculture	complete replacement of fish meat	+300% total ω3 PUFA; +200% PA; +126% EPA; +51,200% DHA; +512% ω3:ω6	[221]
*Schizochytrium* sp.	Feed for Atlantic salmon *Salmo salar* aquaculture	the complete replacement of fish	+6% SFA; +3% DHA; +1% MUFA; +1% total ω6 PUFA	[209]
*Schizochytrium* sp.	Feed for Rainbow Trout *Oncorhynchus mykiss* aquaculture	complete replacement of fish meat	+18% total PUFA	[222]
*Schizochytrium* sp.	Feed for red seabream *Pagrus major* aquaculture	complete replacement of fish meat	+130% SFA; +180% PA; +100% DHA; +2070% DHA:EPA	[223]
*Schizochytrium* sp.	Feed for broiler chicken *Gallus domesticus* farming	0.2% *Schiochytrium* inclusion	+2.5% total USFA	[224]
*Schizochytrium* sp.	Feed for channel catfish *Ictalurus punctatus* aquaculture	+2% dried *Schizochytrium*	+3.71% total ω3 LC-PUFA; +3.35% DHA	[225]
*Schizochytrium* sp.	Feed for shrimps *Litopenaeus vannamei* aquaculture	until 75% replacement of fish meat	+100% total ω6 PUFA; +200% DHA	[226]
*Schizochytrium* sp.	Feed for light lamb *Ovis aries* farming	3.8% *Schizochytrium* and 5% linseed inclusion	+400% ALA; +6500% DHA	[227]
*Schizochytrium* sp.	Feed for rabbit *Oryctolagus* sp. farming	4 gr *Schizochytrium* per kg feed	+50% EPA; +180% DHA	[228]
*Schizochytrium* sp.	Feed for Atlantic salmon *Salmo salar* aquaculture	50% *Schizochytrium* inclusion	+2% PA; +22% total ω6 PUFA; +15% total PUFA; +340% DHA:EPA	[229]
*Schizochytrium limacinum*	Feed for grouper *Epinephelus lanceolatus* aquaculture	in combination with soybean meal, soy protein concentrate could replace 40% of fish meat	+100% DHA; +550% DHA:EPA	[230]
*Schizochytrium limacinum*	General aquaculture feed	48% *v/v* effluent concentration from biofuel industry	+80% SA; +120% DHA	[231]
*Aurantiochytrium* sp.	Feed for black tiger shrimp *Penaeus monodon* aquaculture	1–2% *Aurantiochytrium* inclusion	+200% total MUFA; +10% total SFA; +12% total PUFA; +20% total ω3 PUFA; +37% DHA	[232]
*Isochrysis* sp.	Feed for European seabass *Dicentrarchus labrax* aquaculture	20% of protein and 36% of lipid could be replaced using the freeze-dried *Isochrysis*	+4% total ω3 PUFA; +13% total SFA	[233]
*Isochrysis galbana*	Feed for silverfish *Trachinotus ovatus* aquaculture	24–26% fish oil replacement	+10% DHA; +10% total LC-ω3 PUFA	[234]
*Phaeodactylum tricornutum*	Feed for Atlantic salmon *Salmo salar* aquaculture	6% dried *Phaeodactylum* inclusion	same nutritional content	[215]
*Phaeodactylum tricornutum* or *Crypthecodinium cohnii*	Feed for gilthead seabream *Sparus aurata* aquaculture	2–5% *Crypthecodinium* or *Phaeodactylum* inclusion	+16% total SFA; +20% DHA	[235]
*Haematococcus pluvialis*	Feed for Rainbow Trout *Oncorhynchus mykiss* aquaculture	10 gr *Haematococcus* per kg feed	-	[236]
*Staurosira* sp.	Feed for broiler chicken *Gallus domesticus* farming	7.5% *Staurosira* inclusion	+25% total lipids	[237]
*Arthrospira platensis*	Feed for Nile tilapia *Oreochromis niloticus* aquaculture	30% *Arthrospira* inclusion	-	[238]
*Spirulina platensis*	Feed for broiler chicken *Gallus domesticus* farming	10 gr *Spirulina* per kg feed	same nutritional content	[239]
*Spirulina platensis* or *Chlorella vulgaris*	Feed for African catfish *Clarias gariepinus* aquaculture	ultill 75% *Spirulina* or *Chlorella* inclusion	+20% total ω6 PUFA; +50% DHA	[240]
*Chlorella* sp.	Feed for crucian carp *Carassius auratus* aquaculture	in combination with 2 gr cellulases per kg could completely replace fish meat	-	[241]
*Chlorella* spp.	Feed for Channel Catfish *Ictalurus punctatus* aquaculture	15% *Chlorella* inclusion	+75% PA; +30% OA; +100% EPA; +70% DHA; +32% total ω3 PUFA	[242]
*Chlorella* sp. or *Nannochloropsis* sp.	Feed for European seabass *Dicentrarchus labrax* aquaculture	15% *Chlorella* or *Nannochloropsis* inclusion	+2% total ω6 PUFA	[243]
*Nannochloropsis gaditana*	Feed for Nile tilapia *Oreochromis niloticus* aquaculture	30% *Nannochloropsis* inclusion	-	[244]
*Nannochloropsis gaditana*	Feed for gilthead seabream *Sparus aurata* aquaculture	2% *Nannochloropsis* inclusion	+80% EPA; +200% EPA:DHA	[245]
*Nannochloropsis* sp.	Feed for kuruma shrimp *Marsupenaeus japonicus* aquaculture	4–7–10% *Nannochloropsis* biomass or lipid inclusion	+13% total ω3 PUFA; +44% total ω6 PUFA; +37% EPA	[246]
*Nannochloropsis* sp.	Feed for European seabass *Dicentrarchus labrax* aquaculture	5–10–15% *Nannochloropsis* inclusion	same nutritional content	[247]
*Nannochloropsis salina*	Feed for Nile tilapia *Oreochromis niloticus* aquaculture	complete replacement of fish meat	+47% total ω6 PUFA; +130% EPA	[248]
*Nannochloropsis* sp. or *Pavlova viridis*	Feed for European seabass *Dicentrarchus labrax* aquaculture	complete replacement of fish meat	+50% total PUFA; +70% total ω6 PUFA	[249]
*Pavlova lutheri*	Feed for oyster *Crassostrea gigas* aquaculture	complete replacement of fish meat	-	[250]
*Tetraselmis suecica*	Feed for shrimps *Litopenaeus vannamei* aquaculture	complete replacement of fish meat	-	[251]
*Tetraselmis chuii*	Feed for shrimps *Litopenaeus vannamei* aquaculture	50% *Tetraselmis* inclusion	+2% total lipids	[252]
*Tetraselmis* sp.	Feed for gilthead seabream *Sparus aurata* aquaculture	10% *Tetraselmis* inclusion	same total lipids	[253]
*Tetraselmis suecica* or *Tisochrysis lutea*	Feed for European seabass *Dicentrarchus labrax* aquaculture	combination of both microalgae for a complete replacement of fish meat	-	[254]
*Desmodesmus* sp.	Feed for Atlantic salmon *Salmo salar* aquaculture	20% *Desmodesmus* inclusion	-	[255]
*Aurantiochytrium* sp. and *Schizochytrium* sp.	Feed for gilthead seabream *Sparus aurata* aquaculture	blend of poultry and one of two algal oils	+200% total ω6 LC-PUFA; +3% PA; +500% DHA:EPA	[256]
*Nannochloropsis* sp, *Isochrysis* sp. and *Schizochytrium* sp.	Feed for Rainbow Trout *Oncorhynchus mykiss* aquaculture	blend of three microalgae for a complete replacement of fish meat	+80% total PUFA; +170% total ω6 PUFA	[257]
*Nannochloropsis* sp. and *Schizochytrium* sp	Feed for Nile tilapia *Oreochromis niloticus* aquaculture	Blend of both microalgae for a complete replacement of fish meat	+20% PA; +50% DHA; +20% total ω6 LC-PUFA	[211]

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
