# Peer review of "Highly Valuable Polyunsaturated Fatty Acids from Microalgae: Strategies to Improve Their Yields and Their Potential Exploitation in Aquaculture"

_molecules, 2021, doi:10.3390/molecules26247697_

Round 1

Reviewer 1 Report

Dear authors,

your MS has good quality, both your tables an figures are clear.

The manuscript is well-edited, logically structured and its topic is relevant today.

But I have a few comments:

1a, please, change the formulas of the fatty acids in the full manuscript from type "C18:2 ω-6" to C18:2n-6 or C18:2ω6, but your hybrid forms with different sizes of charachters are incorrect.

1b, please change the name or formula of fatty acids

-from α-linoleic to α-linolenic in the 56th row,

-from C22:6 ω-3 to C22:5n-3 or C22:5ω3 in the 329th row

-from C20:3 ω-6 to "dihomo-γ-linolenic acid – DHGLA, C20:3n-6 or C20:3ω6" in the 424th row

-from γ-linoleic to γ-linolenic in the 432nd row

-from "dihomo-γ-linoleic acid – DGLA, C20:3 ω-6) " to DHGLA in the 445th row

-please insert the formula of ETA in the 385th row

-please insert the formula of DTA in the 395th row

-please change the from LC-ω-6 PUFA  to n-6 LCPUFA or ω6 LCPUFA in the 663th row

 2, The Figure 1 is beautyful, but please, insert the full formula of the all of unsaturated fatty acids (unfortunately, C18:3, C20:4, C22:5 are double). This would make the graph very easy to understand for a researcher not very experienced in fatty acids.

3, please, change from com-pany to company in the 723th row

4, Please check the all of company links in the last column in Table 3.

I needed manually repair the links in the head of my browser in the more cases.  ie from https://www.cyano-tech.com/our-purpose/ to https://www.cyanotech.com/our-purpose/

in fact, at least one case the links need password.

5, the link in the 725th row  This link takes me to the main page, the document is unavilable

6, the link in the 728th and 729th row is wrong

7, The link in the 756th and 757th rows points to the author's computer.

I would like a minor revision.

Author Response

Rev#1

Dear authors,

your MS has good quality, both your tables and figures are clear.

The manuscript is well-edited, logically structured and its topic is relevant today.

But I have a few comments:

1a, please, change the formulas of the fatty acids in the full manuscript from type "C18:2 ω-6" to C18:2n-6 or C18:2ω6, but your hybrid forms with different sizes of charachters are incorrect.

1b, please change the name or formula of fatty acids

-from α-linoleic to α-linolenic in the 56th row,

-from C22:6 ω-3 to C22:5n-3 or C22:5ω3 in the 329th row (line 333)

-from C20:3 ω-6 to "dihomo-γ-linolenic acid – DHGLA, C20:3n-6 or C20:3ω6" in the 424th row (line 451)

-from γ-linoleic to γ-linolenic in the 432nd row

-from "dihomo-γ-linoleic acid – DGLA, C20:3 ω-6) " to DHGLA in the 445th row

-please insert the formula of ETA in the 385th row

-please insert the formula of DTA in the 395th row

-please change the from LC-ω-6 PUFA  to n-6 LCPUFA or ω6 LCPUFA in the 663th row

We thank the referee for these important corrections. Unfortunately, we are not familiar with formulae and chemical nomenclature of fatty acids, since there are no authors with an expertise in organic chemistry. We changed wrong formulae with correct ones.

 2, The Figure 1 is beautiful, but please, insert the full formula of the all of unsaturated fatty acids (unfortunately, C18:3, C20:4, C22:5 are double). This would make the graph very easy to understand for a researcher not very experienced in fatty acids.

we made the suggested corrections

From 3) to 7): We will check point-to-point all the corrections, but, since one of the reviewers suggested us to eliminate the paragraph regarding the current market of microalgae-derived PUFAs, these suggestions will be applied to our forthcoming publication, that will be submitted to a different journal, mainly focused on the techno-economic aspects related to the production of high-valuable products from natural sources. we apologise with rev#1 for spending his time to correct the whole manuscript, including the parts that will be not included in the final version.

3, please, change from com-pany to company in the 723th row

4, Please check the all of company links in the last column in Table 3.

I needed manually repair the links in the head of my browser in the more cases.  ie from https://www.cyano-tech.com/our-purpose/ to https://www.cyanotech.com/our-purpose/

in fact, at least one case the links need password.

5, the link in the 725th row  This link takes me to the main page, the document is unavilable

6, the link in the 728th and 729th row is wrong

7, The link in the 756th and 757th rows points to the author's computer.

I would like a minor revision.

Reviewer 2 Report

The manuscript ID molecules-1507761 entitled “High-valuable fatty acids from microalgae: strategies to improve their yields and their application on global market” is an interesting and valuable study. The Authors carefully analysed the available literature related to the production of fatty acids by microalgae. The use of 268 literature items is very impressive.

I suggest that Chapter 2. Modulation of growth conditions to enhance the production of PUFAs should be supplemented with the results of current optimization studies (2021) presented recently in a series of articles published in the Energies journal: https://doi.org/10.3390/en14061685, https: // doi.org/10.3390/en14102952. The research results presented there directly correspond to the subject of the manuscript ID molecules-1507761 and are a significant supplement to the extensive literature in this field.

The Authors properly present the current scientific achievements in the field of synthesis of these valuable food products by microalgae. The authors effectively try to analyse and summarize the current state of advancement of this type of technology and the possibilities of their application in practice, and on this basis present the potential and prospects of their future use. The manuscript is written in the correct language, its layout and the extracted chapters are logical. The presented content corresponds to the Molecules journal profile.

I believe that the use of microalgae to produce high-valuable fatty acids from microalgae is inextricably linked with the concept of algal biorefinery as a sustainable approach to valorize algal-based biomass towards multiple product recovery. I suggest mentioning this issue in the introduction, because the biomass of microalgae after the recovery of valuable products is used for fertilization, fodder or energy purposes: https://doi.org/10.1016/j.biortech.2019.01.104, https://doi.org /10.1016/j.biortech.2017.01.006, https://doi.org/10.3390/su12239980

I believe that the manuscript may be a valuable work, it is a valuable compendium of scientific knowledge. It also presents limitations and bottlenecks in the possibility of scale-up, which is a great advantage of this study. I believe that it may be published after the additions.

Good luck!

Author Response

Rev#2

The manuscript ID molecules-1507761 entitled “High-valuable fatty acids from microalgae: strategies to improve their yields and their application on global market” is an interesting and valuable study. The Authors carefully analysed the available literature related to the production of fatty acids by microalgae. The use of 268 literature items is very impressive.

I suggest that Chapter 2. Modulation of growth conditions to enhance the production of PUFAs should be supplemented with the results of current optimization studies (2021) presented recently in a series of articles published in the Energies journal: https://doi.org/10.3390/en14061685, https: // doi.org/10.3390/en14102952. The research results presented there directly correspond to the subject of the manuscript ID molecules-1507761 and are a significant supplement to the extensive literature in this field.

We thank the referee for the suggestion. Citing studies on the exploitation of organic wastes as substrate for heterotrophic cultivation of microalgae further improves this manuscript. We added some information in the section 2.5, and cited the articles mentioned by him/her.

The Authors properly present the current scientific achievements in the field of synthesis of these valuable food products by microalgae. The authors effectively try to analyse and summarize the current state of advancement of this type of technology and the possibilities of their application in practice, and on this basis present the potential and prospects of their future use. The manuscript is written in the correct language, its layout and the extracted chapters are logical. The presented content corresponds to the Molecules journal profile.

I believe that the use of microalgae to produce high-valuable fatty acids from microalgae is inextricably linked with the concept of algal biorefinery as a sustainable approach to valorize algal-based biomass towards multiple product recovery. I suggest mentioning this issue in the introduction, because the biomass of microalgae after the recovery of valuable products is used for fertilization, fodder or energy purposes: https://doi.org/10.1016/j.biortech.2019.01.104, https://doi.org /10.1016/j.biortech.2017.01.006, https://doi.org/10.3390/su12239980

The authors agree with the referee: it is of crucial importance to mention the possibility of exploiting algal biomass as source of high valuable commodities and energies, minimizing costs and waste generation. So, according to his/her suggestion, we introduced some sentences in the introduction to better explain this concept. We also cite some relevant papers focused on this topic, including those suggested by him/her.

I believe that the manuscript may be a valuable work, it is a valuable compendium of scientific knowledge. It also presents limitations and bottlenecks in the possibility of scale-up, which is a great advantage of this study. I believe that it may be published after the additions.

Good luck!

Reviewer 3 Report

The manuscript molecules-1507761, High-valuable fatty acids from microalgae: strategies to improve their yields and their application on global market, present a very large amount of data on an important subject. The presentation is multidisciplinary and could be useful for many types of researchers. In my opinion the manuscript seems too long and it would be easier for the reader to understand if the authors would shorten it. In this context, the section 5 could be interesting, but is too far from the scope of the journal. It also seems to be an advertisement for some companies and their products. I appreciate all the work of the authors but I strongly advise them to remove all this section. The authors should prepare another article for economics journal and include this section on it.

I don’t think the Molecules journal readers would be interested in all the companies producing and commercializing microalgae and their websites, but they would be interested in how the fatty acids are extracted and isolated. Are there some specific procedures designed for microalgae? The authors should replace the section 5 with a new section on the subject of extraction and purification of FA. Please detail on the stability of the PUFAs.

The authors should change the title to better represent the paper. The authors discuss almost exclusively about PUFA, and not all fatty acids. The title should reflect the content of the paper.

The introduction could be improved by presenting and commenting on fatty acids effects on health and disease. There are important reviews works on this subject.

The authors should detail in the introduction on the types of fatty acid and better explain the meaning of omega-3 and omega-6 or 9(ω-3, ω-6). Shortly explain the numbers in brackets. The review would considerably benefit if the authors would add a scheme with the structures of α-linolenic acid (ALA), eicosapentaenoic acid (EPA), docosahexaenoic acid (DHA), docosapentaenoic acid (DPA), arahidonic acid, oleic acid, and other FA they are discussing in the paper. Considering the journal, please provide also the chemical names for all the acids mentioned in the introduction (ex. linoleic acid = all-cis-9,12-octadecadienoic acid)

I also think that the paper needs a figure to summarize the main types of microalgae. Just as a model, please take a look at figure 1 from the paper “Microalgae Encapsulation Systems for Food, Pharmaceutical and Cosmetics Applications”

On row 57, it is better to use AA for arahidonic acid.

The authors should add a new section on the stability of PUFA.

There are some editorial mistakes that the authors should have checked before the submission and should be corrected. In table 1, use the point for decimal numbers, and not comma. The presenting style should be the same. I think the system +100% is better than 2X and it should be used in the whole table.

Author Response

Rev#3

The manuscript molecules-1507761, High-valuable fatty acids from microalgae: strategies to improve their yields and their application on global market, present a very large amount of data on an important subject. The presentation is multidisciplinary and could be useful for many types of researchers. In my opinion the manuscript seems too long and it would be easier for the reader to understand if the authors would shorten it. In this context, the section 5 could be interesting, but is too far from the scope of the journal. It also seems to be an advertisement for some companies and their products. I appreciate all the work of the authors but I strongly advise them to remove all this section. The authors should prepare another article for economics journal and include this section on it.

we thank the referee for this important suggestion. We removed the whole section regarding the current market of microalgae-derived PUFAs.

I don’t think the Molecules journal readers would be interested in all the companies producing and commercializing microalgae and their websites, but they would be interested in how the fatty acids are extracted and isolated. Are there some specific procedures designed for microalgae? The authors should replace the section 5 with a new section on the subject of extraction and purification of FA. Please detail on the stability of the PUFAs.

We replaced the section with a new one which fits with the scopes of the journal and of this special issue. We described in this new section the most widespread methods of PUFAs extraction and purification, presenting also the drawbacks related to PUFAs stability, which are of crucial importance to avoid degradation and thus the quality of the final products.

The authors should change the title to better represent the paper. The authors discuss almost exclusively about PUFA, and not all fatty acids. The title should reflect the content of the paper.

We changed the title, but we really appreciate any further suggestion if the referee is not satisfied of the new one.

The introduction could be improved by presenting and commenting on fatty acids effects on health and disease. There are important reviews works on this subject.

We improved the introduction according to the referee’s suggestions, providing more information on health benefits derived from PUFAs.

The authors should detail in the introduction on the types of fatty acid and better explain the meaning of omega-3 and omega-6 or 9(ω-3, ω-6). Shortly explain the numbers in brackets.

Done.

The review would considerably benefit if the authors would add a scheme with the structures of α-linolenic acid (ALA), eicosapentaenoic acid (EPA), docosahexaenoic acid (DHA), docosapentaenoic acid (DPA), arahidonic acid, oleic acid, and other FA they are discussing in the paper. Considering the journal, please provide also the chemical names for all the acids mentioned in the introduction (ex. linoleic acid = all-cis-9,12-octadecadienoic acid)

We thank the referee for this important suggestion, and we prepared a new table according to his/her instructions.

I also think that the paper needs a figure to summarize the main types of microalgae. Just as a model, please take a look at figure 1 from the paper “Microalgae Encapsulation Systems for Food, Pharmaceutical and Cosmetics Applications”

We inserted a new figure. Unfortunately, we did not have the possibility of inserting some pictures of the algal species, since we have no time to ask for copyright permission. But, in our opinion, the figure seems clear and improves the quality of the manuscript, since it contains information about both divisions/classes and the name of the species/genera which are mainly used to produce PUFAs.

On row 57, it is better to use AA for arachidonic acid.

Done.

The authors should add a new section on the stability of PUFA.

Done. As suggested in the second comment, we provided details on PUFAs stability in the new section.

There are some editorial mistakes that the authors should have checked before the submission and should be corrected. In table 1, use the point for decimal numbers, and not comma. The presenting style should be the same. I think the system +100% is better than 2X and it should be used in the whole table.

We corrected the errors in the table, standardizing the method used to present the data.

Round 2

Reviewer 2 Report

Great work has been done. Congratulations !!!

Author Response

thanks for the positive evaluation of the manuscript, and for helping us to improve it!

Reviewer 3 Report

The authors performed all the changes suggested in the first review. The manuscript was considerably improved. I only suggest that the authors should move the figure 2 at the start of the paper. I think it would be best to present it before section 2. It would be easier for the readers that are not very familiar with all the species presented in sections 2 and 3.

Author Response

we thank the referee for this further suggestion. we inserted the figure 2 (nov become fig. 1) in the first section of the manuscript.